# Role of charges in a dynamic disordered complex between an IDP and a folded domain

Katrine Bugge [1] ✉, Andrea Sottini [2], Miloš T. Ivanović [2], Freia S. Buus[1], Daniel Saar [1], Catarina B. Fernandes[1], Fabienne Kocher [2], Jacob H. Martinsen[1], Benjamin Schuler [2,3] ✉, Robert B. Best [4] ✉ & Birthe B. Kragelund [1] ✉

Protein complexes involving intrinsically disordered proteins (IDPs) cover a continuum from IDPs that fully fold upon binding to IDPs that remain fully disordered in the complex. Here we demonstrate a case of charge-driven interactions of a folded domain with an oppositely charged IDP that remains completely disordered in the complex. Using the negatively charged and fully disordered prothymosin α and the positively charged and folded globular domain of histone H1.0, we show that they form a low-micromolar-affinity complex without fixed relative orientations or persistent contacts between specific residues. Using 25 charge variants of the globular domain, we find that the binding affinity can be modulated both by net charge and charge clustering on the folded domain, indicating some selectivity in highly charged complexes. Our results highlight that a folded protein can provide a charged surface onto which an oppositely charged IDP can bind while retaining disorder. We expect that more such complexes exist.

Intrinsically disordered proteins (IDPs) and regions (IDRs) exist in ensembles of disordered configurations in their functional form[1-5]. Due to their large surface exposure, IDPs have a rich binding capacity for other macromolecules, and several recent studies have shown how IDPs expand molecular communication through, e.g., folding upon binding, multispecificity, and conformational buffering, fostering a rethinking of protein interaction modes[6-11]. The mechanisms by which IDPs bind their partners have expanded both in number and complexity, and there are now several examples of complexes in which IDPs retain a high level of disorder that is important for their biological functions[12-16]. In many of these cases, the IDPs tend to have a high charge density and low-complexity sequences[17], providing them with properties resembling polymer chains and allowing highly charged IDPs to behave as flexible polyelectrolytes[18].

Although some examples of dynamic hydrophobic interactions have been seen, e.g., in kinase scaffolding by the IDR of the Na$^+$/H$^+$-exchanger 1[19], the FG-Nups[20] or in the complex between E-cadhedrin and β-catenin[21], biomolecular interactions where IDPs retain dynamic disorder in complexes are often driven by electrostatics[18]. Here, multiple residues of opposite charge drive the interaction and allow for structural disorder to be retained, without the need for complementary structured interfaces or folding upon binding. Electrostatically driven dynamic interactions have been shown to assist short linear motifs (SLiMs) when these fold upon binding into hydrophobic pockets with remaining disorder in the charged flanking regions. Examples include the LxxIxE SLiM binding to the protein phosphatase PP2A[22] and the QxxLxxFF SLiM binding to proliferating cell nuclear antigen, PCNA[23]. However, dynamic interactions between opposite charges can also be more specific, as in the case of multisite phosphorylations in Sic1[24], where a single phosphoryl group binds into a specific positively charged binding pocket. Because of the dynamics in the complex, adding additional phosphoryl groups increases the

[1]REPIN and the Structural Biology and NMR Laboratory, The Linderstrøm-Lang Centre for Protein Science, Department of Biology, University of Copenhagen, Copenhagen, Denmark. [2]Department of Biochemistry, University of Zurich, Zurich, Switzerland. [3]Department of Physics, University of Zurich, Zurich, Switzerland. [4]Laboratory of Chemical Physics, National Institute of Diabetes and Digestive and Kidney Diseases, National Institutes of Health, Bethesda, MD, USA. ✉e-mail: katbugge@gmail.com; schuler@bioc.uzh.ch; robert.best2@nih.gov; bbk@bio.ku.dk

binding affinity allovalently[24–26], enhancing affinity through a local concentration effect of the phosphoryl groups. An extreme example of a disordered protein complex involves the two oppositely and highly charged IDPs prothymosin α (ProTα) and linker histone H1.0 (H1)[16]. Although ProTα and H1 interact with picomolar to nanomolar affinity at physiological ionic strengths, they fully retain their disorder without formation of persistent contacts between specific residues[16], and irrespective of stereochemistry[27]. However, despite the emerging insight into the importance of charges in the dynamic interactions by IDPs, the nature of interactions between polyelectrolytic IDPs and highly charged folded protein domains remains unexplored. Furthermore, the roles of the number and distribution of charges for binding and selectivity have not yet been experimentally addressed in a systematic way.

ProTα is a multifunctional, polyanionic IDP that can serve as a model polyelectrolyte. It contains no folded domains and is highly negatively charged (net charge of -44 (or -43 depending on isoform[7]), with 49% of its sequence constituted by Glu or Asp[28]), with most of the charges clustered around the center and towards the C-terminus (Fig. 1A). ProTα is involved in many biological functions, including chromatin remodelling[29], transcription[30], cellular proliferation[31], oncogenesis[32] and apoptosis[33], binding several different interaction partners to exert its functions. In chromatin remodeling, the partner protein is the linker histone H1. H1 has a positive net charge of +53 and consists of a small globular domain (GD)[34] flanked by two disordered tails (Fig. 1A). In isolation, GD has similar properties in terms of structure and stability as in the context of full-length H1[34]. H1 plays a major role in chromatin condensation and transcriptional regulation[35–37], but also in oncogenesis[38,39]. The chromatin-condensing properties of H1 are mainly conferred by its long polycationic C-terminal tail[40,41], interacting with and condensing inter-nucleosomal linker DNA[41,42]. To remodel chromatin, ProTα binds to H1 and acts as a chaperone, by extracting H1 from the nucleosome and increasing the mobility of H1 in the nucleus[41,43]. Recent work has shown that the inherent dynamics of such complexes facilitates the formation of short-lived ternary complexes that lead to the rapid exchange of binding partners by competitive substitution[44], and thus to concentration-dependent ligand exchange kinetics[7,41]. The affinity between ProTα and full-length H1 is dominated by the disordered regions[16], but the GD of H1, a small 70-residue folded domain carrying an overall positive net charge of +9, still contributes to binding with its low micromolar affinity for ProTα[16]. The nature of the interaction between ProTα and the folded GD is unresolved.

In the present work, we use GD and ProTα as models for investigating the interaction between a polyelectrolytic IDP and a highly charged and folded protein domain. We find that ProTα retains its disorder in the complex, without the formation of structure, fixed relative orientations, or persistent contacts between specific residues. Using 25 single-, double- and quadruple amino acid substitutions in GD, we systematically modulate its overall net charge and surface charge clustering. We find that net charge has the dominant effect on the affinity for ProTα, but that charge clustering also matters. Thus, polyelectrolyte interactions are strongly influenced by the number of charges and their surface clustering, suggesting that selectivity in polyelectrolyte interactions may be encoded by these features.

## Results

### ProTα remains disordered in complex with a folded partner

To understand the nature of the interaction between the fully disordered ProTα and the surface of the folded GD, we used solution-state NMR spectroscopy (Fig. 1). Addition of GD to $^{15}$N-ProTα induces chemical shift perturbations (CSPs) of a large fraction of the ProTα backbone amide resonances, albeit of modest amplitude (Fig. 1C, H). In the complex, the overall peak dispersion of ProTα resonances remains similar to the unbound state (Fig. 1C). Strikingly, the NMR peaks, and

hence the CSPs, move on the same path, irrespective of whether ProTα was titrated with the H1-GD or full-length H1 (Fig. 1D), but with substantial differences in amplitude (Fig. 1H). Usually, different protein partners result in unique "fingerprints" of the chemical shifts of the observed protein. However, as the binding between ProTα and H1 is primarily driven by electrostatics without site-specific residue-residue interactions[16], the difference in amplitude resulting from the addition of either H1-GD or full-length H1 may be explained by differences in the average density of proximate charges. While the amplitude of the CSPs of ProTα saturates near equimolar addition of full-length H1[16], addition of H1-GD does not fully saturate the CSP amplitudes, even at 1·8x molar ratio of H1-GD (Fig. 1B). This is consistent with the lower net charge of H1-GD ( + 9) compared to full-length H1 ( + 53).

Comparing the perturbations of the NMR backbone relaxation parameters of $^{15}$N-ProTα upon addition of H1-GD with those upon addition of full-length H1 recorded at 750 MHz (Fig. 1E–G) and at 800 MHz[16], reveal similar patterns, albeit with differences in amplitudes. As with H1, the longitudinal ($R_1$) relaxation rates are almost unperturbed by complex formation (Fig. 1E). The transverse ($R_2$) relaxation rates and the heteronuclear $^{15}$N-{$^1$H} nuclear Overhauser effects (hetNOEs) are, however, modestly perturbed within the most acidic region of ProTα, the same region displaying the largest CSPs (Fig. 1F, G). The modest increase in both $R_2$ (average of 3.1 s$^{-1}$ to 3.5 s$^{-1}$ at 750 MHz with 4xGD) and hetNOEs (average of 0.12 to 0.15 with 4xGD) is consistent with a small retardation of the backbone dynamics for ProTα in complex with GD, nonetheless still well within the range observed for fully disordered chains[16]. Finally, to assess whether binding to GD induces the formation of secondary structure in ProTα, we assigned the $^{13}$C-chemical shifts of the backbone nuclei of ProTα at equimolar ratio of GD and at full saturation with GD (Supplementary Fig. 1). The secondary chemical shifts (SCSs) were unperturbed by binding of GD, as was the case for full-length H1[16], underscoring the absence of structure induced in ProTα.

Together, these findings suggest that ProTα engages with the folded and net positively charged surface of H1-GD in a similarly dynamic manner as with the mainly disordered full-length H1. The proteins thus bind to each other without the formation of secondary or tertiary structure, and without structurally well-defined interaction sites or fixed relative orientations of the two proteins.

### ProTα binds the charged surface of GD without a well-defined binding site

To characterize the interaction from the GD perspective (Fig. 2A), we titrated $^{15}$N-GD with unlabeled ProTα to concentrations ~10 times the $K_D$,[16] and analyzed perturbations of NMR observables. In the $^1$H-$^{15}$N-HSQC spectra, the resonances of GD exhibited fast to intermediate exchange, with some line broadening for Tyr28, Ala43, Lys69, and Ser71 (Supplementary Fig. 2). The resulting amide CSPs (Fig. 2B) are broadly distributed in the primary and tertiary structure of GD rather than clustered in any specific region (Fig. 2B, E). No consistent correlations were found between CSPs and charged sidechains or solvent accessible surface areas (SASA) (Fig. 2B), although the largest CSPs were observed for residues located on the GD surface, suggesting no structural rearrangement upon interaction (Fig. 2B, E and Supplementary Fig. 2). The GD CSP amplitudes increased up to the addition of roughly two times molar ratio of ProTα, after which near-saturation was reached (Fig. 2C).

To quantify the affinity between ProTα and GD, we used single-molecule Förster resonance energy transfer (smFRET) spectroscopy[45]. Picomolar concentrations of ProTα labeled at positions 56 and 110 with Alexa Fluor 488 and Alexa Fluor 594, respectively, were incubated with increasing concentrations of GD up to 100 μM. The transfer efficiency ($E$) histograms (Fig. 1I) revealed that the mean transfer efficiency, ⟨$E$⟩, which is related to ProTα compactness, increases continuously with increasing GD

 

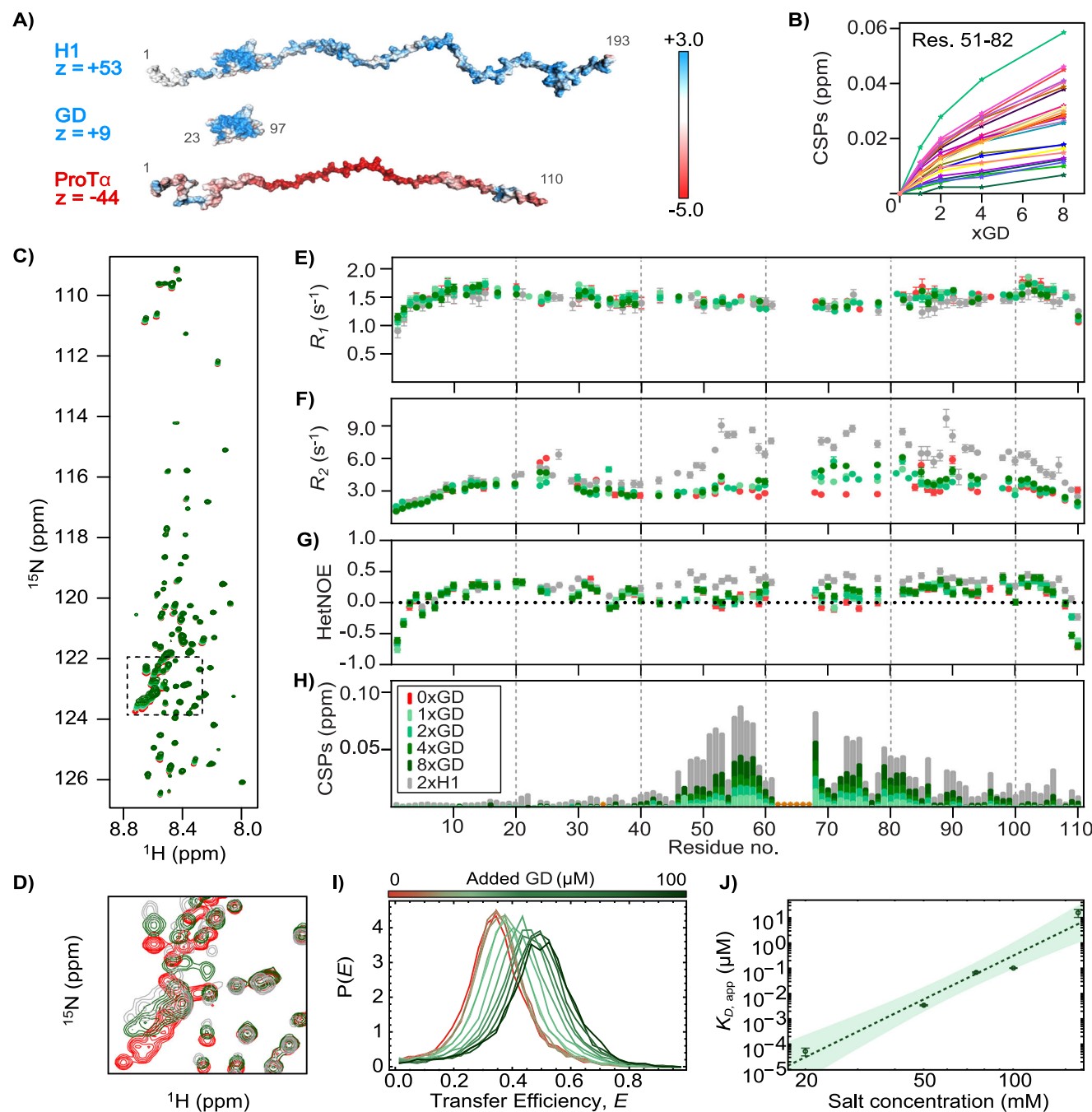

**Fig. 1 | ProTα remains disordered and dynamic in complex with H1-GD.**
**A** Illustrations of ProTα, full-length H1 (H1), and H1-GD (GD), with net charges (z), and surface electrostatic potentials in red-white-blue color scale (units: $k_BT$ per e⁻) (partially reproduced from Borgia et al.[16]). **B** Backbone CSPs (free ProTα − complex ProTα) of 28 μM $^{15}$N-ProTα (residues 51-82, individual colors) plotted against times molar ratio GD relative to the free state (max: 8xGD (224 μM); lines drawn to guide the eye). In (**C**–**H**) full-length H1 or different molar ratios of H1-GD added to $^{15}$N-ProTα (see color key in **H**); orange stars: unassigned/missing data or insufficient data quality. **C** $^1$H-$^{15}$N-HSQC spectra of $^{15}$N-ProTα titrated with GD. **D** Zoomed-in region of the $^1$H-$^{15}$N-HSQC spectra of $^{15}$N-ProTα, with 8x molar ratio of H1-GD, and 2x molar ratio of full-length H1 (corresponding to dashed box in **C**)). **E** $R_1$ $^{15}$N-relaxation rates, **F** $R_2$ $^{15}$N-relaxation rates, and (**G**) HetNOE values of $^{15}$N-ProTα. Data in (**E**–**G**)

were recorded at 750 MHz on 100 μM $^{15}$N-ProTα with times molar ratio of H1-GD as indicated by legend, or of 37 μM $^{15}$N-ProTα with 74 μM full-length H1 (under these conditions, both dimers and trimers are populated[7,46], but their relaxation behavior is very similar[16]). **H** CSPs of $^{15}$N-ProTα with full-length H1 or H1-GD at different molar ratios. **I** Transfer efficiency histograms of fluorescently labeled ProTα with increasing concentrations of H1-GD at 165 mM ionic strength. **J** Plot of $K_{D,app}$ for 1:1 binding of H1-GD to ProTα as a function of ionic strength, fitted using the Lohman-Record theory[47] (shaded area: 90% confidence interval). Similar data to those presented in (**B**, **C**, **H**) have been published in the supplemental data of Borgia et al. 2018[16], but at different molar ratio. Data in (**B**, **I**, **J**) reported in Source Data file. Errors are stadard errors from the fit.

concentration. This observation is indicative of fast exchange between the expanded conformational ensemble of free ProTα and the more compact ones in complex with GD during diffusion through the confocal volume on the millisecond timescale. This

fast exchange observed by smFRET is consistent with the fast exchange and the relaxation data observed by NMR (Fig. 1C–G).

The fast exchange observed both by NMR and smFRET complicates the population analysis by concealing the possible presence of

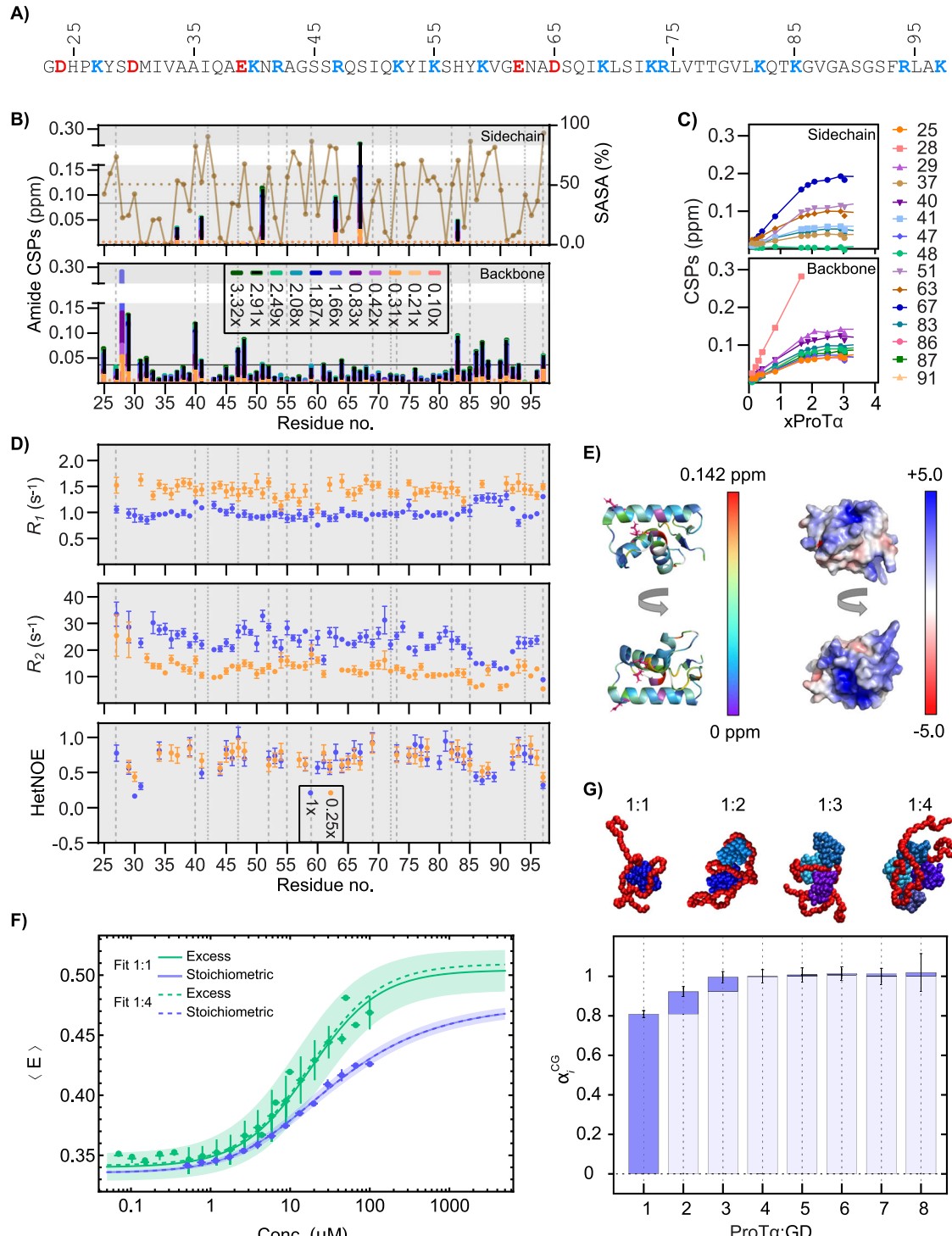

**Fig. 2 | GD binds ProTα primarily in a 1:1 complex without distinct binding sites.** **A** GD sequence with acidic and basic residues colored. **B** CSPs of sidechain (top) and backbone (bottom) amides of $^{15}$N-GD (54 μM) with different molar ratios of ProTα. Top: %SASA (brown). Red asterisks: unassigned/missing data, horizontal lines: average SASA. Vertical lines: Lys (dashed), Arg (dotted). **C** GD (54 μM) CSPs versus times molar ratio of ProTα (up to 3.32x) relative to the free state. Curves represent individual residues (lines to guide the eye). **D** Perturbation of $^{15}$N-GD at different GD:ProTα ratios: Top: $R_1$ $^{15}$N-relaxation rates; middle: $R_2$ $^{15}$N-relaxation rates; bottom: HetNOEs (600 MHz; 185 μM $^{15}$N-GD and 184 μM ProTα, or 40 μM $^{15}$N-GD with 0.25x molar ratio of ProTα). Data for free $^{15}$N-GD[34] shown in Supplementary Fig. 4A. **E** Cartoon representation of GD (PDB code: 6hq1[34]) from two angles colored by CSP magnitude (color gradient, left), white residues: no data: pink: sidechains with CSPs >average. Right: GD with surface electrostatic potentials in red-white-blue color scale. **F** Mean transfer efficiency, ⟨E⟩, of donor/acceptor-

labeled free ProTα (⟨E⟩₀ = 0.334 ± 0.008) and increasing concentrations of GD in 165 mM ionic strength buffer. Picomolar labeled ProTα titrated with unlabeled GD (green), and with increasing equimolar concentrations of unlabeled ProTα and GD (blue). Continuous lines and shaded areas are fits and their standard errors, respectively, assuming a 1:1 ProTα:GD binding model. Dashed lines: fit to 1:4 binding model. We note that the analysis approach based on the insights developed here yields $K_D$ values ~ one order of magnitude greater than reported[16] (see Methods). **G** Illustrations of complexes between ProTα (red) and increasing number of GDs (blue/purple) from coarse-grained simulations. The normalized relative increase in ⟨E⟩ of ProTα in the different complexes, $\alpha_i^{GD} = (\langle E\rangle_i^{CG} - \langle E\rangle_0^{CG})/(\langle E\rangle_4^{CG} - \langle E\rangle_0^{CG})$, calculated from the coarse-grained simulation. Data in Fig. (**B**–**D**, **F**, **G**) reported in Source Data file. Errors are standard errors from the fit.

different oligomers. At lower ionic strength, however, the exchange rate between free ProTα and its GD-bound states decreases, and we can identify up to four subpopulations in the transfer efficiency histograms (Supplementary Fig. 3). The population and depopulation of the peaks observed with increasing GD concentration suggest that each peak corresponds to a subpopulation with increasing binding stoichiometry of GD to ProTα, analogous to the behavior of full-length H1 binding to ProTα identified previously[46]. Moreover, the subpopulations observed with increasing GD concentration exhibit increasing transfer efficiency, as expected from a compaction of ProTα with an increasing number of oppositely charged GD molecules bound. Overall, this result strongly suggests that multiple GD molecules can bind to one ProTα chain, with decreasing affinity for higher oligomers because of the anti-cooperativity resulting from the smaller number of charged groups in ProTα effectively available per copy of GD bound (Supplementary Fig. 3). From the results, we thus estimated the affinities between ProTα and GD in the corresponding subpopulations.

To be able to quantify the binding mechanism also at higher ionic strength, where the subpopulations cannot be separated, we utilized a coarse-grained model[16] to simulate the interaction of an increasing number of GD molecules with ProTα at an ionic strength of 165 mM. These results are in line with the conclusion that ProTα can populate complexes with GD of different stoichiometries (Fig. 2F, G). When simulated in the presence of twenty GD molecules, ProTα spends most of its time in a 1:4 complex, where the charges between the two partners are almost balanced (−44 vs. +36 ( = + 9·4)). The binding of a single GD in the simulations increases $\langle E \rangle$ of ProTα in the complex to ~80% compared to $\langle E \rangle$ for ProTα saturated with four GD molecules (Fig. 2G). The binding of further GD molecules increases $\langle E \rangle$ only by ~11% and ~7% in the 1:2 and 1:3 complexes, respectively (Fig. 2G). To estimate the impact of multiple GD molecules binding to ProTα on the experimental data, we fitted $\langle E \rangle$ as a function of GD concentration with either a 1:1 or a 1:4 binding model, where in the latter, the relative increase in $\langle E \rangle$ was fixed to the values obtained from the coarse-grained simulations (Fig. 2F). A similar analysis was conducted for stoichiometric titrations (blue points in Fig. 2F) at 165 mM ionic strength, where the contribution of stoichiometries with more than one GD molecule bound to ProTα was minimized by measuring increasing concentrations of an equimolar ratio of ProTα and GD. The results of the fits (Fig. 2F) with the 1:1 and 1:4 models demonstrate that both models describe the data well, but the 1:4 model is overparameterized: only the $K_D$ for formation of the 1:1 complex with GD and its transfer efficiency report meaningful values and uncertainties (Supplementary Table 1). Moreover, these parameters yielded similar values in both analyses (Fig. 2F and Supplementary Table 1), i.e., with an excess or equimolar amounts of GD relative to ProTα. These observations suggest that at an ionic strength of 165 mM, the majority of ProTα compaction in complex with GD results from the formation of the 1:1 complex, and the observed dissociation constant, $K_{D,app}$, is dominated by the formation of the 1:1 complex. Note that the coarse-grained simulations used here did not explicitly consider counterions. While counterion release is an important phenomenon in the binding of charged proteins[46], we have shown that this model, which considers ions implicitly via screening of coulombic interactions, is sufficient to reproduce the structural ensembles of bound complexes of charged proteins[16].

The fits of $\langle E \rangle$ as a function of GD concentration yield apparent affinities of the 1:1 complex between ProTα and GD of $K_{D,app} = 17 \pm 6\,\mu M$ (excess titration) and $15 \pm 2\,\mu M$ (stoichiometric titration) at 165 mM ionic strength. By analyzing the affinity of the 1:1 complex as a function of ionic strength with the Lohman-Record formalism[47] (Fig. 1J and Supplementary Fig. 3), we found that $4.6 \pm 0.6$ counter ions are released upon binding, compared to ~18 counter ions for the interaction of ProTα with H1[16]. This strong dependence of the affinity on salt concentration reveals the pronounced electrostatic

contribution to the interaction between ProTα and GD, and the difference in the number of counter ions released reflects the greater contribution of the disordered regions of H1 to the H1:ProTα affinity compared to the GD.

We next asked whether the broad range of binding stoichiometries would affect the dynamics of the complex as monitored by NMR. In the 1:1 stoichiometric mixture, the $R_1$ and $R_2$ relaxation rates of GD are almost uniformly perturbed upon addition of ProTα, whereas the hetNOEs are essentially unperturbed (Fig. 2D and Supplementary Fig. 4A). $R_1$ decreases from an average of $1.7\,s^{-1}$ to $1.0\,s^{-1}$ from free GD to equimolar addition of ProTα, while $R_2$ increases from an average of $9.5\,s^{-1}$ to $23.5\,s^{-1}$. These changes likely originate from slower tumbling of the structured GD upon interaction with the disordered ProTα. Accordingly, the uniform distribution of $R_1$ and $R_2$ perturbations along the chain, the lack of changes in hetNOEs together with increased scattering in the $R_2/R_1$ versus $R_1 \cdot R_2$ plot (Supplementary Fig. 4B), point towards changes in $\tau_c$, without any localized effects on GD. In comparison, in the 4:1 stoichiometric mixture, the $R_1$ and $R_2$ relaxation rates of GD are barely perturbed upon addition of 0.25-fold molar addition of ProTα, $R_1$ decreases from an average of $1.7\,s^{-1}$ to $1.4\,s^{-1}$, and $R_2$ increases from an average of $9.5\,s^{-1}$ to $13.7\,s^{-1}$. Similarly, only minor differences were seen when evaluated in the $R_2/R_1$ and $R_1 \cdot R_2$ plots (Supplementary Fig. 4B). Assuming the formation of a 4:1 complex between GD and ProTα, the GDs compete for interaction with ProTα, resulting in each GD having fewer contacts with ProTα on average, compared to GD in a 1:1 complex. If only the 1:1 complex formed, the contributions from free GD would likely dominate the data, leading to similar results.

Since changes in chemical shifts and in $R_1$ and $R_2$ relaxation rates could be observed across the entire sequence of GD in complex with ProTα, not one specific but several regions of GD are involved in the interaction with ProTα Combined, these observations are consistent with a charge-driven complex without a specific binding interface, allowing the formation of complexes of various stoichiometries.

## ProTα forms a dynamic complex with GD without persistent contacts

To obtain an atomistic picture of the complex and its dynamics, we performed all-atom molecular dynamics simulations with explicit solvent, which have the potential to capture both the conformational ensemble as well as the relevant time scales for the dynamics of the complex (Fig. 3). The simulations were first validated against the experimental data by comparing the experimental NMR relaxation parameters for ProTα with those computed from the simulation (Fig. 3B). Overall, the results are in good agreement considering the difficulty of sampling these interactions in simulations: there is agreement of the magnitude and residue-to-residue variation of $R_1$, $R_2$ and hetNOEs between simulation and experiment. Just as important, the qualitatively small changes in these parameters in going from unbound to bound states are also reproduced in the simulations (Fig. 3B, C). The regions in which there is the greatest disagreement with experiment, e.g., $R_2$ for residues 50-80 with one GD bound, are also the regions with the largest statistical errors. NMR relaxation parameters for GD were also in reasonable agreement with experiment (Supplementary Fig. 5).

The simulations revealed that the complex is highly dynamic (Supplementary Movie 1), as evident in the distribution of intermolecular residue-residue contact lifetimes with a mean lifetime of $4.5 \pm 0.6\,ns$ (Fig. 3C). The correlation time for fluctuations in the radius of gyration of ProTα, a measure of global chain relaxation time, was just 35 ns. This timescale is close to the chain reconfiguration time of $42_{-1}^{+8}$ ns inferred from nanosecond fluorescence correlation measurements[48,49] of FRET-labeled ProTα (Supplementary Fig. 6), further supporting the validity of the simulations. The combined simulation and experimental results thus suggest that ProTα stays in contact with GD by constantly breaking and forming different

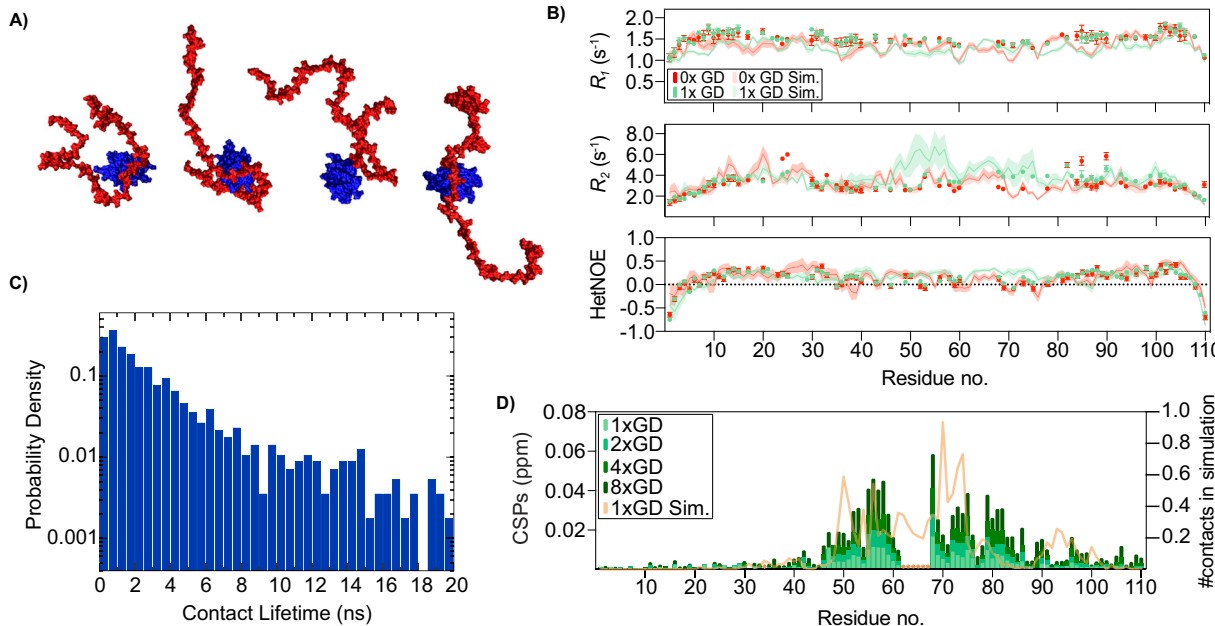

**Fig. 3 | Fast timescale dynamics is maintained in the 1:1 ProTα-GD complex.**
**A** Examples of configurations of the PoTα (red) and GD (blue) complex from all-atom molecular dynamics simulations (see also Supplementary Movie 1). **B** Comparison of experimentally determined relaxation rates of $^{15}$N-ProTα alone and with equimolar addition of GD, measured at 750 MHz, and relaxation rates determined from simulations (see color key). Errors from the simulations are shown as shaded areas and error bars are standard errors of the fit to experimental data. **C** Probability density for contact lifetimes. **D** Overlay of CSPs of ProTα induced by 1-8x molar ratio of GD (see color key) and the average number of contacts of ProTα to GD in the 1:1 simulation (orange line) based on the simulations. Data in Fig. 3B reported in Source Data file.

contacts, without any specific long-lived interface. As a final validation of the all-atom simulation results, we have compared the number of residue-residue contacts formed by each residue of ProTα with CSPs on binding, both of which reflect the regions of ProTα to which the GD most frequently binds (Fig. 3D). The qualitative consistency of these two measures suggests that the GD is binding in the same region in the simulations as in the experiment.

It is important to point out that the results were dependent on the protein force field used. Several alternative force fields yielded a complex that was apparently too tight, with strongly elevated $R_2$ values for ProTα in the complex that were inconsistent with the experimental results (Supplementary Fig. 7 and Supplementary Table 2). It appears that these elevated $R_2$ values are the result of persistently formed salt bridges[50], resulting in slower global dynamics not compatible with experiment and with a tail of long-lived contacts for all force fields except for the des-amber force field specifically modified to avoid this artifact[51] (Supplementary Fig. 8). This sensitivity of the relaxation parameters to the dynamics of the ProTα:GDinterface supports our conclusion that the complex is stabilized by many short-lived interactions: if the alternative model is having long-lived contacts between the two proteins, we would expect to see strongly elevated $R_2$ values, which is not observed experimentally.

**Net charge is not the only determinant of affinity**
As the interactions between H1-GD or H1 and ProTα are predominantly electrostatically driven, and since the affinity of ProTα for GD (net charge of +9) is much weaker than that for H1 (net charge of +53), this suggests an affinity dependence on partner net charge. To test this hypothesis, we first asked whether ProTα would interact with two non-related charged proteins; the folded RST domain of the plant protein RCD1[52] with a net charge of +5, and the folded human calmodulin (CaM) with a net charge of −24 (Fig. 4A). Surprisingly, addition of eight times molar ratio of RST to $^{15}$N-ProTα not only induced CSPs of the NMR resonances of ProTα, but the paths and the CSP-per-residue

pattern were similar to those from additions of H1 and H1-GD, only with lower amplitudes (Fig. 4B, C and Supplementary Fig. 9). Hence, RST interacts with ProTα, and the fingerprint of the interaction supports a similar interaction as with H1 and H1-GD. Addition of 8x molar ratio of the negatively charged CaM, however, did not result in any detectable CSPs, suggesting no interaction. Hence, net charge is an important factor for binding ProTα.

To systematically investigate the role of net charge for affinity, we constructed ten variants of GD with different net charge, from +9 for GD-WT to +5 (2), +7 (3), +11 (3) and +13 (2)[34] (Table 1; parentheses show number of variants with the same net charge). This was done by combinations of (i) replacing different Lys residues with Gln, (ii) substituting different uncharged, solvent exposed residues with Lys, and (iii) replacing different Asp or Glu residues with Asn or Gln[34]. We previously confirmed that the GD structure was unperturbed by these mutations, while concluding that a net charge of +13 is the limit to keep GD folded[34]. All variants with a net charge of +7 or higher induced similar resonance trajectories and CSP-per-residue patterns on ProTα (Fig. 4D and Supplementary Fig. 10), whereas the perturbations imposed by the +5 variants were hardly detectable. However, the CSP amplitudes relative to the amplitudes at the same concentration of GD-WT differed between the net charge variants; the higher the net charge, the higher the amplitude (Fig. 4D and Supplementary Fig. 10).

The affinities of ProTα for the GD charge variants were quantified by smFRET (Supplementary Fig. 11), where the reported apparent affinities are for the 1:1 complexes. These were inferred using the fitting procedures described above for the GD-WT (Fig. 2) and assuming that the fundamental behavior as a function of salt concentration, especially the dominance of the first binding event in the transfer efficiency change, does not differ between GD-WT and the GD variants. No change in transfer efficiency was observed for the two +5 net charge variants up to a concentration of 100 μM, while the apparent $K_D$ values of the +7 to +13 net charge variants ranged from 191 ± 54 μM to 1.3 ± 0.2 μM (Fig. 4E and Table 1), respectively. As suspected, the apparent $K_D$ values show a clear trend with net charge: The higher the

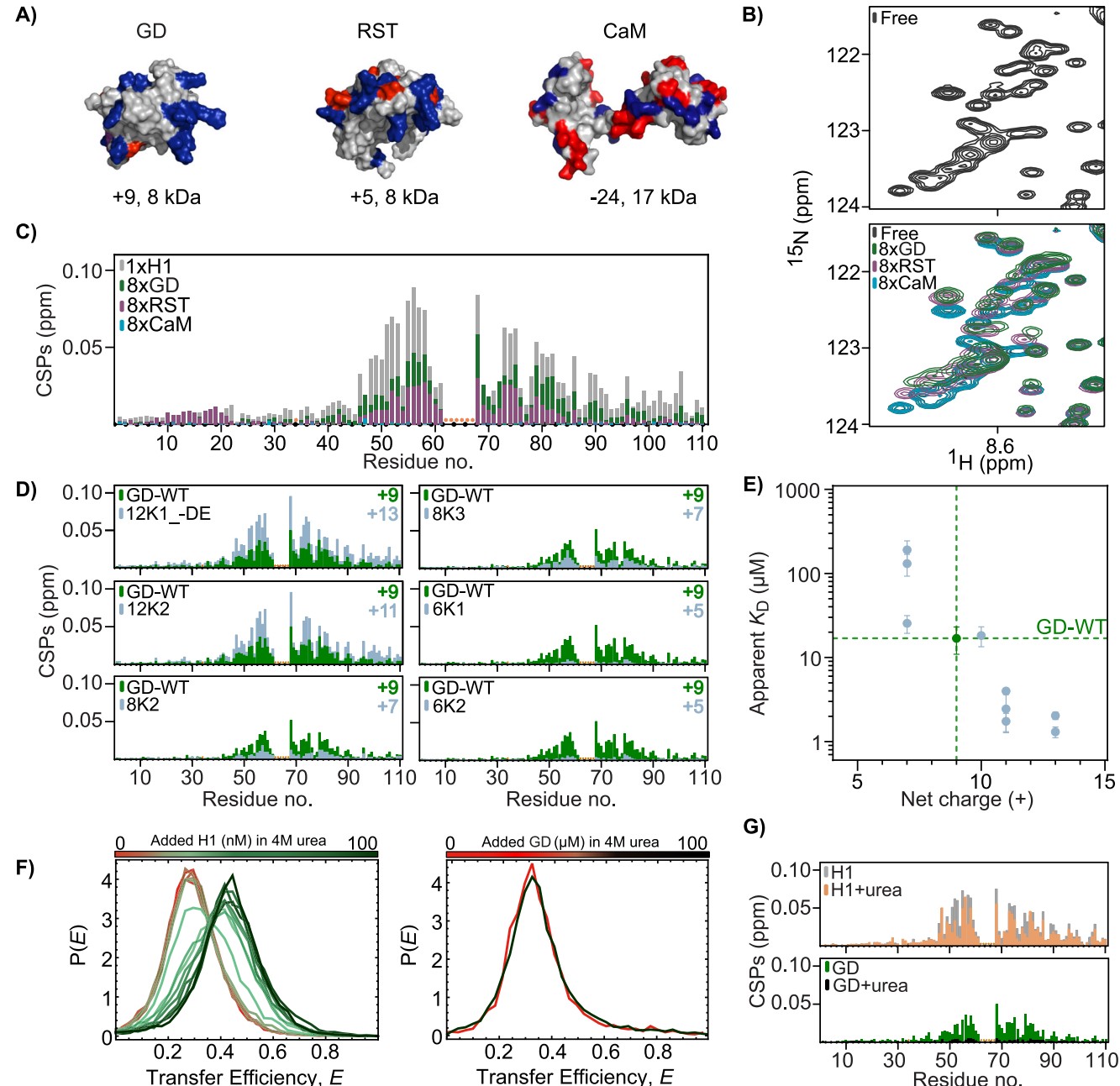

**Fig. 4 | Net charge is a major affinity determinant. A** Molecular surfaces of GD (PDB code: 6hq1[34]), RCD1-RST (PDB code: 5oao[52]) and CaM (PDB code: 1cll[96]) with basic residues in blue and acidic residues in red, and net charge and molecular weight indicated below. **B** Region of the $^{15}$N-HSQC spectra of $^{15}$N-ProTα (28 μM) in its free state (black) on top and same region below with 8x molar ratio of either GD (green), RST (purple) or CaM (light blue). **C** CSPs of $^{15}$N-ProTα (28 μM) amides at 8x molar ratio of GD (green), RST (purple) or CaM (light blue), compared to CSPs at 2x full-length H1 (grey), plotted against residue number. **D** CSPs of $^{15}$N-ProTα (50 μM) upon addition of 4x molar ratio of either GD-WT (green) or GD charge variants (blue grey, see Table 1) plotted against residue number. Orange asterisks: unassigned residues. Additional data in Supplementary Fig. 10. **E** Apparent $K_D$ plotted against net charge for GD-WT (green dashed lines) and ten different GD charge variants (light blue). The error is the standard deviation of $K_D$ from individual fits of $n = 3$. **F** Transfer efficiency histograms of labeled ProTα binding to full-length H1 (left) and to H1-GD-WT (right) in the presence of 4M urea and at an ionic strength of 200 mM and 165 mM, respectively. **G** Comparison of CSPs of $^{15}$N-ProTα (25 μM) at 2x molar ratio of full-length H1 and at equimolar addition of full-length H1 in the presence of 4M urea (top) or 4x molar ratio of H1-GD-WT with or without the presence of 4 M urea (bottom). Data in (**C**, **D**, **F**, **G**) reported in Source Data file.

positive net charge, the lower the apparent $K_D$ (Fig. 4E). A difference of +6 in net charge changed the apparent $K_D$ by a factor of 100, suggesting that net charge is a major determinant for the interaction strength between ProTα and binding partners. Nonetheless, we observed some distribution in the apparent $K_D$ values among variants with the same net charge; the apparent $K_D$ values of the three +7 net charge variants range from 191 ± 54 μM to 26 ± 6 μM, hinting towards other contributions to affinity than net charge alone. This conclusion

was further substantiated by comparing the interaction of folded and urea-unfolded GD with ProTα, a comparison that allows keeping the amino acid composition and net charge constant while changing their relative positions and hence charge clustering. Compared to urea-unfolded H1, which readily binds ProTα[16], unfolded GD (Supplementary Fig. 12) had no detectable affinity for ProTα in 4M urea (Supplementary Fig. 13) in smFRET experiments (Fig. 4F) and induced only minute CSPs on $^{15}$N-ProTα (Fig. 4G). Altogether, this suggests that the

**Table 1 | GD variants**

| Variant name | Substitutions | Net charge | $T_m$ (K) | Relative CSP$_{sum}$ | Apparent $K_D$ (µM) |
|---|---|---|---|---|---|
| WT | - | +9 | 321.8 ± 0.1 | 1.0 ± 0.1 | 17 ± 6 |
| 4R4K | R42K, R47K, R74K, R94K | +9 | 318.5 ± 0.1 | 0.80 ± 0.08 | 15 ± 8 |
| 12K1 | A34K, Q67K | +11 | 313.4 ± 0.1 | 1.7 ± 0.2 | 2 ± 1 |
| 12K1_DE | A34K, Q67K, D30N, E39Q | +13 | 301.6 ± 0.3 | 2.0 ± 0.2 | 2.0 ± 0.2 |
| 12K2 | L70K, S90K | +11 | 316.5 ± 0.1 | 1.8 ± 0.2 | 4.0 ± 0.2 |
| 2E2K | E39K, E62K | +13 | 313.1 ± 0.1 | 1.8 ± 0.2 | 1.3 ± 0.2 |
| 2E2Q | E39Q, E 62Q | +11 | 313.3 ± 0.1 | 1.4 ± 0.1 | 1.7 ± 0.4 |
| H57K | H57K | +10 | 315.6 ± 0.1 | 1.0 ± 0.1 | 18 ± 5 |
| H57Q | H57Q | +9 | 312.2 ± 0.2 | 1.1 ± 0.1 | 17 ± 2 |
| 8K1 | K82Q, K52Q | +7 | 323.3 ± 0.1 | 0.63 ± 0.06 | 26 ± 6 |
| 8K2 | K85Q, K73Q | +7 | 326.4 ± 0.1 | 0.34 ± 0.03 | 191 ± 54 |
| 8K3 | K59Q, K69Q | +7 | 325.9 ± 0.1 | 0.39 ± 0.04 | 131 ± 38 |
| 6K1 | K82Q, K52Q, K85Q, K73Q | +5 | 327.8 ± 0.1 | 0.11 ± 0.01 | - |
| 6K2 | K85Q, K73Q, K40Q, K97Q | +5 | 325.7 ± 0.1 | 0.13 ± 0.01 | - |
| 73_34 | K73A, A34K | +9 | 322.2 ± 0.1 | 0.54 ± 0.05 | 36 ± 18 |
| 73_66 | K73S, S66K | +9 | 322.1 ± 0.1 | 0.88 ± 0.08 | 17 ± 4 |
| 73_70 | K73L, L70K | +9 | 324.3 ± 0.1 | 1.0 ± 0.1 | 10 ± 2 |
| 74_34 | R74A, A34R | +9 | 317.3 ± 0.1 | 0.63 ± 0.06 | 64 ± 17 |
| 74_67 | R74Q, Q67R | +9 | 318.9 ± 0.1 | 0.99 ± 0.09 | 24 ± 6 |
| 74_70 | R74L, L70R | +9 | 322.6 ± 0.1 | 1.2 ± 0.1 | 5.7 ± 0.3 |
| 82_78 | K82T, T78K | +9 | 321.7 ± 0.1 | 1.3 ± 0.1 | 10 ± 4 |
| 85_90 | K85S, S90K | +9 | 320.0 ± 0.1 | 1.0 ± 0.1 | 10 ± 6 |
| 94_90 | R94S, S90R | +9 | 317.3 ± 0.1 | 1.4 ± 0.1 | 4.6 ± 0.6 |
| 2S1 | K85S, S90K, K73A, A34K | +9 | 319.2 ± 0.1 | 0.57 ± 0.05 | 34 ± 8 |
| 2S2 | R74L, L70R, K73S, S66K | +9 | 321.1 ± 0.1 | 1.0 ± 0.1 | 13.8 ± 0.8 |
| 2S3 | K73Q, R74Q, Q37K, Q48R | +9 | 326.6 ± 0.1 | 1.1 ± 0.1 | 18 ± 3 |

position of the charges relative to each other on the surface, i.e., the degree of charge clustering, is a relevant affinity determining parameter.

**Charge clustering increases affinity**

To systematically address how charge clustering affects interaction strength, a second set of GD variants was designed and produced. In this group of 12 variants, referred to as 'swap variants', the positions of one (9 variants) or two (3 variants) positively charged side chains were swapped with another side chain on the GD surface, conserving the amino acid composition and the net charge (Fig. 5A and Table 1). The single-swap variants can be grouped in two; one group where charges were moved onto, or from, the folded α-helix 3 (α-variants), and one group where charges were moved within the disordered and highly positively charged patch in the β-hairpin loop region (β-variants) (Fig. 5A). In these designs (see also Methods), swaps were restricted to surface-exposed positions and confirmed to be non-disruptive to the structure of GD by CD spectroscopy (Supplementary Fig. 14 and Table 1). In this way, we systematically changed the charge clustering on the GD surface to either obtain more distributed charges, or increase the charge clustering in specific areas, while keeping the net charge constant. In addition, we investigated the effect of Arg and His as cationic residues (Table 1). For all variants, their interaction with ProTα was mapped by NMR and smFRET.

All swap variants induced CSPs in $^{15}$N-ProTα with similar patterns along the sequence, but with different amplitudes (Supplementary Figs. 15, 16). The apparent $K_D$ values of the 1:1 complex with the swap variants, obtained using smFRET and model fitting as described above for GD-WT, ranged from 4.6 ± 0.6 µM (94_90; moving charge from position 94 to position 90) to 64 ± 17 µM (74_34) (Table 1 and Fig. 5B), i.e., a 20-fold difference between specific single swaps. Some swaps

increased the apparent $K_D$ relative to GD-WT (74_67, 2S1, 73_34, 74_34), some decreased it (94_90, 74_70), and some were neutral unchanged (2S2, 85_90, 82_78, 73_70, 73_66, 2S3). In line with the electrostatic nature of the interaction, the 4R4K and the His variants had no effects on $K_D$ (Fig. 5B). The variant 2S1 combines 85_90 and 73_34, which individually had no effect on (85_90), and increased the apparent $K_D$ (73_34), respectively. The apparent $K_D$ of 2S1 was consistent with the sum of the two single-swap variants. For 2S2, which combines 74_70 and 73_66, a less pronounced effect on the apparent $K_D$ than for 74_70 was observed (Fig. 5B). In variants where charges were swapped within the same α-helix (α-swaps) as a group, a small local rearrangement of moving a charge one or two helix turns had no or only modest effects on the apparent $K_D$ (from 5.7 ± 0.3 µM to 24 ± 6 µM). The observation that the effect on affinity is modest and occurs in both directions supports an interaction independent of geometrically well-defined binding sites. Completely delocalizing the charge from the α-helix-2 region to the opposite, charge-depleted region of the structure (73_34 and 74_34) resulted in an increase in apparent $K_D$ to 36 ± 18 µM and 64 ± 17 µM, respectively. When comparing this result to the magnitudes of the apparent $K_D$ values obtained for net charge variants, it corresponds to effectively eliminating the charge for interaction. All variants where charges were swapped within the β strands (β-variants) had no effects on $K_D$, except for 94_90, which reduced the apparent $K_D$ three-fold (5.7 ± 0.3 µM). There was no pronounced correlation between the apparent $K_D$ and the melting temperature $T_m$ for the swap variants (Supplementary Fig. 14C).

The complementarity of the experimental observables is substantiated when correlating the apparent $K_D$ values obtained from smFRET with the normalized CSP$_{sum}$ from NMR. A strong correlation between the CSP amplitudes and the affinity was found (Fig. 5C and Supplementary Figs. 17, 18), suggesting that the amplitude of the CSPs

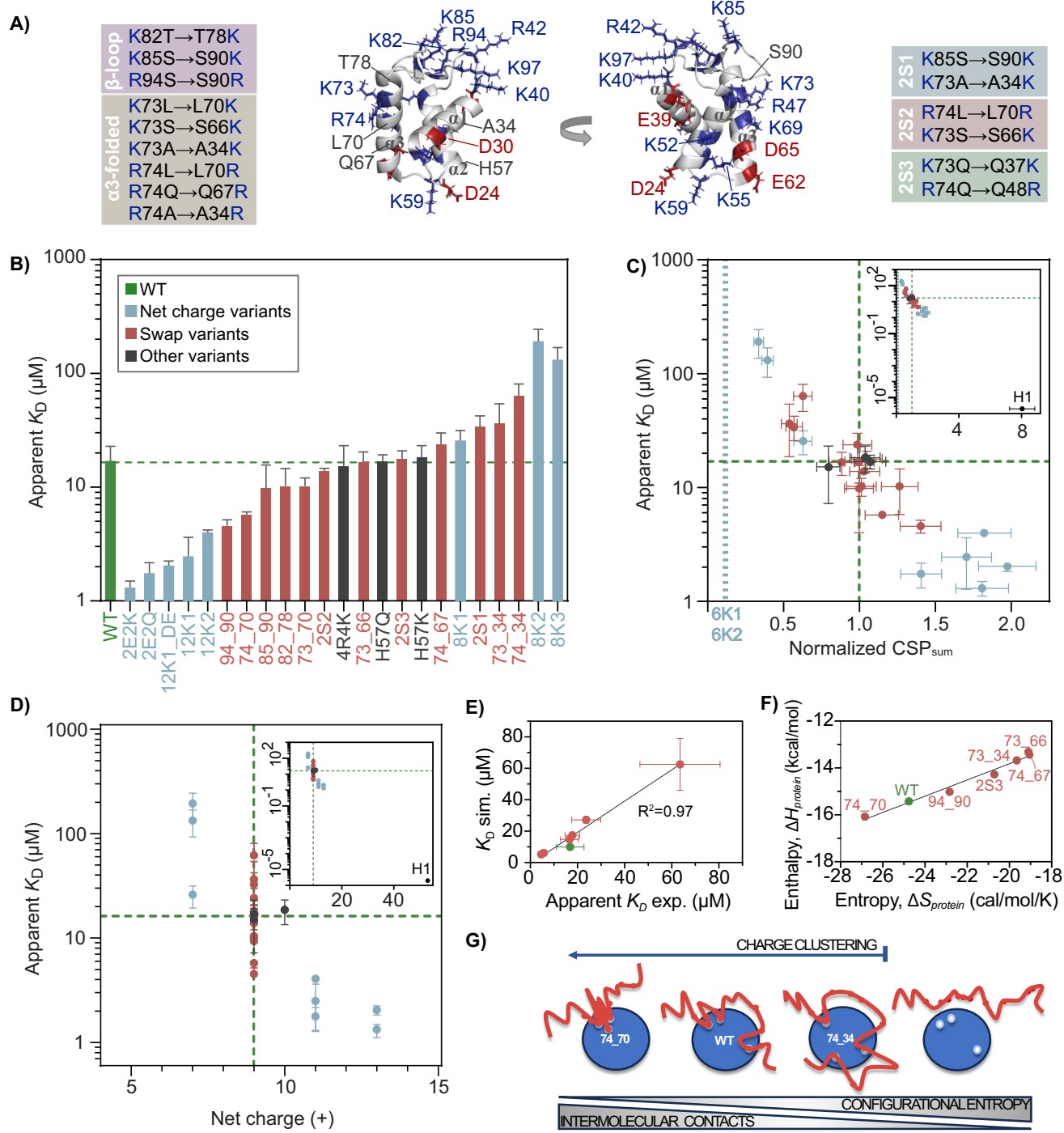

**Fig. 5 | Charge clustering modulates affinity. A** Overview of the 12 charge swap variants of GD. **B** Apparent $K_D$ values for ProTα binding GD-WT and 23 GD variants (**see** Table 1). **C** Apparent $K_D$ plotted against normalized $CSP_{sum}$ calculated at 4x molar ratio of GD (see Methods) for GD-WT and 23 GD variants. Normalized $CSP_{sum}$ of 6K1 and 6K2 (for which no binding was detected by smFRET) are shown as dotted blue lines, while GD-WT values are highlighted by green dashed lines. Insert includes data for full-length H1. Errors are propagated errors from three repetitions of GD-WT. **D** Apparent $K_D$ for ProTα:GD-variants plotted against net charge for 23 GD variants. Dashed green lines highlight result for ProTα:GD-WT. **E** Correlation between $K_D$ from coarse-grained simulations versus $K_D$ measured by smFRET. **F** Correlation between enthalpy ($\Delta H_{Protein}$) and entropy ($\Delta S_{protein}$) of binding for seven of the +9 charge variants and GD-WT from coarse-grained simulations. **G** Schematic illustration of the enthalpy-entropy compensation driven by charge clustering. Data in (**C**, **D**, **F**, **G**) reported in Source Data file. The error on the experimentally determined $K_D$ is the standard deviation of $K_D$ from individual fit of $n = 3$.

is a measure of affinity in this type of complex. Including data from the complex with full-length H1 with a much higher affinity for ProTα[16] is consistent with this conclusion. Correlating apparent $K_D$ values with the overall net charge supports the conclusion that the largest changes in affinity result from changing the net charge compared to differences

in charge clustering (Fig. 5D and Supplementary Fig. 19A). An order-of-magnitude change in apparent $K_D$ roughly requires a change of three in net charge. For some swap variants, relocating a single charge while keeping the net charge constant has a greater effect on affinity than reducing the net charge by two. Hence, both overall net charge and

charge clustering are affinity determinants for this type of disordered dynamic complex between a highly charged IDP and a highly charged folded protein. Finally, to address whether the changes in charge density or net charge of the GD would affect the dynamics of ProTα, we recorded $^{15}$N $R_2$ relaxation rates of ProTα in complex with 4x molar ratio of either GD-WT or a GD charge variant. Here we selected two swap variants with increased (74_70) or decreased (74_67) charge density, respectively, and a variant with increased net charge (2E2Q). For all variants, and similar to GD-WT, we observed an increase in $R_2$ relaxation rates in the binding regions of ProTα, and although the effect was small, the increase was more pronounced for the higher charge density variant 74_70 and the increased net charge variant 2E2Q (Supplementary Fig. 19B). These data agree with the $R_2$-profile of ProTα in complex with the more charged full-length H1[16].

To rationalize how the charge distribution causes the changes in the affinity, we illustrated the surface charge distribution of seven different GD variants including the WT, which all had the same overall net charge of +9 (Supplementary Fig. 20) and ran coarse-grained simulations. In these simulations we were able to recapitulate the distribution of affinities measured experimentally, with good agreement between the apparent $K_D$ values measured by smFRET and those extracted from the simulations (Fig. 5E). From the simulations we then extracted the protein contributions to the entropy ($\Delta S_{protein}$) and enthalpy ($\Delta H_{protein}$) of binding (Fig. 5F) from the temperature-dependence of $K_D$. A classical enthalpy-entropy compensation was apparent[53,54], such that the more contacts formed between ProTα and GD, i.e., the more favorable the binding enthalpy, the larger the loss of entropy. A feature of our model is that it neglects thermodynamic contributions from solvent and ions. We can therefore identify the change in entropy in the model as arising from the protein configurational entropy change on binding. In the context of our model, the differences in the entropy of binding between different GD variants are expected to be dominated by conformational (or configurational) entropy. Interestingly, the variants where the charges were redistributed to generate more charge clustering had more contact energy and a larger loss in entropy, and vice versa for variants where charges became more distributed. This observation suggests that when charges cluster, it is simpler for ProTα to make more favorable Coulombic interactions than in the case when the charges are distributed more uniformly, with a consequent loss of configurational entropy (Fig. 5G).

## Discussion

In this work, using a combination of biophysical and computational methods, we show that a negatively charged IDP can form a micromolar-affinity complex with the surface of a positively charged folded domain while remaining fully disordered and without forming persistent contacts or specific relative orientations. This type of complex expands the spectrum of disordered complexes involving IDPs[5,18,55,56]. To address the specificity and determinants of this type of complex, we use a set of proteins with different net charge and designed a large set of charge variants of the folded GD modulating the overall net charge as well as charge clustering. We found that ProTα can bind different positively charged folded proteins in a similar manner, and that the overall net charge of the partner is a major determinant of the affinity. We also found that the distribution of the charges, the charge clustering, plays an important role, and changes in charge clustering can modulate the affinity in both directions. All variants that increased the apparent $K_D$ led to a disruption of charged patches (e.g., for the net charge variants and for 73/74_34, 2S1). For all variants with increased affinity, the charges were more clustered. Thus, and in agreement with a recent computational study of charge variations of ProTα and H1[53], the relative position of charges matters. The more positive charges that are present on the surface of GD, and the more clustered they are, the larger the loss in configurational entropy of ProTα in the complexes. Importantly, our findings suggest that it is

not the specific position of a charge, as it would be within a traditional binding site, it is its contribution to an overall local and global charge density that is relevant.

Interactions that are independent of local side chain orientations and geometries allow for encoding affinity – and hence selectivity – for disordered partners that is insensitive to rearrangements of ordered binding sites. This is exemplified by H1 and ProTα. In mammals, a multitude of tissue-specific H1 isoforms exist in which the GD is responsible for binding to the DNA on the nucleosome, mainly through geometrically well-defined binding sites. Small variations in positioning of specific charged residues (e.g., H1.0 versus H1.1. or H1.5 versus H1.10[34]) involved in this interaction result in variation in the orientation of the GD in the chromatosomes[57]. This in turn gives rise to different levels of condensation of the nucleosome as different affinities of the histones for chromatin[58]. As a result, different H1 isoforms give rise to differential gene expression[59]. While such subtle, local repositioning of charges can have a large effect on the ordered interaction with DNA, the properties (net charge and charge clustering) that allow the chaperone ProTα to extract H1[41] and hence GD, remain unperturbed.

The role of net charge and charge patterning for the function of IDPs is gaining increased attention with computational approaches important for efficiently screening many sequences with modulated net charge and charge patterning. Several studies have shown that increasing charge clustering promotes chain compaction[17,60–62], modulates properties of polyelectrolyte condensates leading to more extended conformations[63,64], steers intersegmental transfer-efficiency for DNA-bound transcription factors[65] and affects binding affinities, shown for the fully disordered complex between full-length H1 and ProTα[16,53,63]. From these mostly computational studies, it appears that charge densities in IDPs may determine properties of their protein complexes including their condensates such as specificity, affinity, dynamics, and shape. Here we experimentally reveal the magnitude of such modulations and show that surface charge density and charge patterning of a folded protein can also affect these properties in complex with a charged IDP.

Since the binding affinity for this type of complex appears to be encoded in the net charge as well as in charge clustering, these properties, and their variation, may encode yet-to-be deciphered specificity rules between highly charged proteins. Besides sequence variation, protein modifications including phosphorylation[66], acetylation[67] and ubiquitylation[68] modulate overall net charge and charge density and thus become highly relevant as affinity tuners for these types of complexes. We find that high affinity requires high net charge and high charge clustering and thus packing of more charges on a folded protein surface. Negatively charged inner membrane surfaces represent high charge density surfaces that may form similar dynamic complexes with disordered, highly positively charged proteins, as exemplified by the MARCKS proteins[69,70] and the N-terminal region of ChiZ[71]. In these cases, changes in lipid composition e.g., via (de)-phosphorylation of inositides, enable affinity modulation. For the complex between GD and ProTα, we found that changing the GD surface net charge by +6 changed the affinity for ProTα by two orders of magnitude; an effect that could similarly be obtained by doubling or halving the ionic strength. The larger ionic strength sensitivity for the much more charged H1:ProTα complex, where doubling the ionic strength reduces the affinity by six orders of magnitude[16,46], suggests that these types of complexes may be regulated both by posttranslational modifications and by changes in local ion concentrations. Finally, in the absence of charge matching between two polyelectrolytes, different stoichiometries, as we observe here, can occur, with differences in the corresponding affinities. This behavior may be relevant in defining specificities between charged proteins and can play roles in their regulation. Whether there is a corresponding effect of charge clustering in the disordered ProTα on binding is an interesting avenue for future investigation.

Although the dynamic complex between the disordered ProTα and the folded GD adds to the many ways by which both folded and disordered proteins can interact, examples of more weakly formed disordered electrostatic interactions involving a folded and a disordered domain have been reported. In these cases, however, they exist in the context of the same peptide chain. They represent examples of transcription factors[72,73] where a negatively charged IDR is linked to its own positively charged DNA binding domain (DBD) in *cis*, and transcriptional regulators and calcium binding proteins with disordered tails[74,75]. Here, the IDR makes transient and weak interactions with the folded domain, likely enhanced by the direct tethering increasing the local concentration[76], and in some cases leading to a stabilization of the folded domain[75]. Stronger intramolecular interactions have also been observed, as exemplified by the highly dynamic, but high-affinity intramolecular interaction between D/E repeats and the folded domain within HMGB1, leading to dynamic autoinhibition[77]. Although these examples represent an intramolecular effect and have not yet been described in as much detail as here, they highlight the broader presence of highly dynamic interactions between disordered and folded protein domains of biological relevance.

While biologically relevant disordered complexes of the type discovered here exist, it is not clear which biological roles they play. If network formation is achievable from joining several domains, disordered interactions can promote condensation, as seen for ProTα and H1[64], and may support certain dynamic properties of the condensate. Importantly, fast regulation from keeping the IDP disordered in the complex with the folded partner allows for invasion of more chains, as was illustrated here for GD and earlier for H1[7], leading to enhancement of the rate of partner exchange via competitive substitution[7,44]. The disorder and dynamics in the complex thus enable augmented access for competing binding partners despite high affinity and for modifying enzymes that enhance or weaken the interaction by modulating net charge and charge clustering. More examples are needed to fully reveal the functional implications of dynamic complexes between IDPs and folded domains.

## Methods

### Protein production

ProTα and isotope labelled ProTα, GD and isotope labelled GD, GD variants, RST, and CaM were produced as previously described[16,34,52,78]. For NMR, isoform 2 of ProTα (P06454-2) was used, whereas for smFRET, isoform 1 (P06454-1) was used. The two isoforms differ in the core acidic region by one negative charge, negligible for affinity[7]. Unlabeled recombinant wild-type human histone H1.0 for smFRET was from New England Biolabs (cat. # M2501S) or, for NMR, produced recombinantly[16]. For generating the charge swap variants of GD, the following strategy was taken. The single-swap variants overall group in two; one group where charges are moved from the folded α-helix 3 (α-variants), while the other group involves moving charges from the disordered and highly positively charged patch in the β-hairpin loop region (β-variants) (Fig. 5A). In the six single α-variants, the positively charged side chain at either position 73 or 74 was moved one or two turns within the helix or moved to α-helix 1 (A34), which contains a positive charge-depleted patch (Fig. 5A). Furthermore, in the double swap variant 2S3, residues K73 and R74 were relocated to other charged patches. For the β-variants, three charged side chains were repositioned locally (Fig. 5A). The remaining two double-swap variants, 2S1 and 2S2, combined two single-swap variants. In one variant, we substituted all Arg to Lys (4R4K), in one variant His57 with Gln, and in one His57 with Lys.

### Preparation of fluorescently labeled ProTα

Among the variants that have previously been tested for binding, ProTα E56C/D110C, which probes the more highly charged region of

ProTα and was used here, has been shown to exhibit the largest changes in transfer efficiency and is thus the most sensitive to binding events[7,16]. The plasmid encoding the ProTα E56C/D110C variant was transfected into E. coli C41 DE3 cells grown in Terrific Broth medium with kanamycin (50 μg/ml), and protein expression induced by 0.5 mM isopropyl-β-D-1-thiogalactopyranoside (IPTG) at an $OD_{600}$ of -0.7 over night at 25 °C. Cell pellets were collected and resuspended in denaturing buffer (6 M guanidinium chloride (GdmCl) in phosphate-buffered saline (PBS) pH 7.4 with 2 mM dithiothreitol (DTT)); the soluble fraction was collected and applied to a Ni Sepharose Excel resin (Cytiva). The resin was washed twice with 5 resin volumes of denaturing buffer including 25 mM imidazole before applying the extracted sample. The protein was eluted with PBS including 500 mM imidazole and then dialyzed against 50 mM Tris buffer pH 8 + 200 mM NaCl, 2 mM DTT and 1 mM EDTA using a 3.5 kDa molecular cut-off membrane. The hexahistidine tag was cleaved during dialysis using HRV 3C protease. The protein was run through the Ni Sepharose Excel resin once again to remove the cleaved tag, and the flow-through was concentrated using Vivaspin 20 3 kDa molecular weight cut-off concentrators (VIVAproducts). The protein was further purified using a HiPrep-Q FF column for ion exchange chromatography (Cytiva). The column was equilibrated with 50 mM Tris buffer pH 7.4, 200 mM NaCl and 2 mM DTT, and after loading the protein on the column, ProTα was eluted in 50 Tris pH 7.4 with a gradient from 200 mM to 1 M NaCl. Fractions containing the purified protein were collected and concentrated before being buffer exchanged using a HiTrap Desalting column (Cytiva) against freshly prepared and degassed labeling buffer with 100 mM potassium phosphate at pH 7. The eluted protein was labeled by incubating it with 0.7:1 Alexa Fluor 488 dye to protein molar ratio for 1 h at room temperature and sequentially with 1.5:1 Alexa Fluor 594 fluorophore to protein molar ratio over night at 4 °C. Finally, the labeled protein was purified first by using the HiTrap Desalting column and then by reversed-phase high-performance liquid chromatography (RP HPLC) on a SunFire C18 column (Waters Corporation) with an elution gradient from 20% (v/v) acetonitrile and 0.1% (v/v) trifluoroacetic acid in aqueous solution to 37% acetonitrile. ProTα-containing fractions were lyophilized and stored at -80°C.

### Circular dichroism (CD) spectropolarimetry

Far-UV CD spectra were recorded using a Jasco-J-815 spectropolarimeter installed with a Peltier controlled cuvette holder. All spectra were recorded at 10 °C between 260 and 195 nm, data pitch was 0.5 nm and a digital integration time of 2 s, path length of 0.1 cm and a scan speed of 50 nm/min, accumulating 10 scans, used. Only measurements at a dynode voltage below 700 V were included, and identical settings were used to record a spectrum of the buffer which was then subtracted. The proteins were dissolved in TBS buffer, pH 7 at room temperature (20–21 °C) at a concentration of 20 ± 0.5 μM. The ellipticity was converted to mean residue weight ellipticity using Eq. (1).

$$[\theta]_{MRW} = \frac{\frac{MW}{(n-1)} mdeg}{10\, c\, d} \tag{1}$$

where $[\theta]_{MRW}$ is the mean residue weight ellipticity, $c$ the concentration in g/L, $n$ the number of residues, $d$ the path length in cm and MW the molecular weight in Da.

The chemical stability of GD-WT was determined using urea denaturation. Far-UV CD spectra were recorded at different urea concentrations ranging from 0 M to 7 M urea. The data pitch was 0.2 nm, the digital integration time was set to 2 s, path length was 1 mm, a bandwidth of 1 nm, scanning speed of 20 nm/min with 10 accumulations. From monitoring the change in mdeg at 222 nm, the

unfolding curve was fitted with Eq. (2).

$$y = \frac{(a_N[urea] + b_N) + (a_D[urea] + b_D)\exp\left(\frac{m([urea]-c_m)}{RT}\right)}{1 + \exp\left(\frac{m([urea]-c_m)}{RT}\right)} \quad (2)$$

where y is the observed signal, $a_N$ and $a_D$ are the intercepts of the baselines with the y-axis before and after the transition, respectively, [urea] is the concentration of urea, $b_N$ and $b_D$ are the slopes of the baselines before and after transition, respectively, $c_m$ is the concentration of urea at the denaturation midpoint, $R$ is the gas constant and $T$ is the absolute temperature.

To determine thermal stability, melting experiments were performed from 283 to 353 K in increments of 1 K/min, monitoring the ellipticity change at 222 nm. Ellipticity was sampled every 0.1 °C, and the sample was allowed to return to the start temperature after which a spectrum was recorded for assessing reversibility. The thermal melting curves were fitted with Eq. (3).

$$y = \frac{(m_N T + y_N) + (m_D T + y_D)\exp\left(-\frac{\Delta H\left(1-\frac{T}{T_m}\right)}{RT}\right)}{1 + \exp\left(-\frac{\Delta H\left(1-\frac{T}{T_m}\right)}{RT}\right)} \quad (3)$$

where $y$ is the observed signal, $y_N$ and $y_D$ are the pre- and post-transition baseline intercepts, respectively, $m_N$ and $m_D$ are the corresponding slopes of the baselines, $\Delta H$ is the van't Hoff enthalpy at $T_m$, $T$ is the temperature, $T_m$ is the melting temperature. Errors reported in Table 1 are standard errors of the fit. For calculating the change in stability, $\Delta\Delta G_{N\text{-Dapparent}(WT\text{-}MUT)}$, we made the assumptions that $\Delta C_p$ is temperature-independent and that the changes in $\Delta C_p$ are close to zero. We used the following equation[79] to calculate $\Delta\Delta G_{N\text{-}D}$ apparent (at 298 K) (Eq. 4):

$$\Delta\Delta G_{N-D\,apparent}(T) \approx \frac{-\Delta H_{average}^{Tm} T}{(T_m^{WT} T_m^{Mut})}\Delta T_m \quad (4)$$

where $\Delta H_{average}^{Tm}$ is the average folding enthalpy at $T_m$, $T_m^{WT}$ and $T_m^{Mut}$ are the melting temperatures of WT and protein variant, respectively, and $\Delta T_m$ is the difference in melting temperature between the WT and the protein variant ($\Delta T_m = T_m^{mut} - T_m^{WT}$).

## Free-diffusion single-molecule FRET

smFRET experiments were conducted using either a custom-built or a MicroTime 200 confocal microscope (PicoQuant, Berlin, Germany) equipped with a 485-nm diode laser and an Olympus UplanApo 60x/1.20 W objective. After passing through a 100 µm pinhole, sample fluorescence was separated into donor and acceptor components using a dichroic mirror (585DCXR, Chroma, Rockingham, VT). After passing appropriate filters (Chroma ET525/50M, HQ650/100), each component was focused onto avalanche photodiodes (SPCM-AQR-15, PerkinElmer Optoelectronics, Vaudreuil, QC, Canada), and the arrival time of every detected photon was recorded (Hydra Harp 400, PicoQuant, Berlin, Germany). The 485-nm diode laser was set to an average power of 100 µW (measured at the back aperture of the objective), either in continuous-wave or pulsed mode with alternating excitation of the dyes, achieved using pulsed interleaved excitation (PIE). The wavelength range used for acceptor excitation in PIE mode was selected with a z582/15 band pass filter (Chroma) from the emission of a supercontinuum laser (EXW-12 SuperK Extreme, NKT Photonics) driven at 20 MHz, which triggered interleaved pulses from the 485 nm diode laser used for donor excitation. In our experiments, photon bursts originating from an individual molecule diffusing through the confocal volume (at least 3000 bursts) were selected against the background mean fluorescence counts and, in the case of pulsed interleaved excitation, by a stoichiometry ratio S of 0.2<S<0.75.

Transfer efficiencies were quantified according to $E = n_A/(n_A + n_D)$, where $n_D$ and $n_A$ are the numbers of donor and acceptor photons in each burst, respectively, corrected for background, channel crosstalk, acceptor direct excitation, differences in quantum yields of the dyes, and detection efficiencies[80].

All data analysis was conducted using the Mathematica (Wolfram Research) package Fretica (https://schuler.bioc.uzh.ch/programs). All smFRET experiments were performed in µ-Slide sample chambers (Ibidi, Germany) at 22 °C in TEK buffer[16] with varying KCl concentrations of 20–160 mM (the ionic strengths quoted throughout the manuscript include the 8 mM ionic strength of 10 mM Tris at pH 7.4); 140 mM 2-mercaptoethanol and 0.01% (v/v) Tween-20 were added for photoprotection and for minimizing surface adhesion, respectively. In experiments with excess GD, donor/acceptor-labeled ProTα was added to a final concentration of 50–100 pM to ensure single molecule conditions, while unlabeled GD was added at different concentrations, up to 100 µM. In stoichiometric titration experiments, the equimolar ratio of ProTα and GD was favored by using an increasing concentration of a 1:1 molar ratio of unlabeled ProTα and GD, up to 100 µM, while labeled ProTα was kept at a concentration of 50–100 pM.

## Analysis of binding affinities

Transfer efficiency histograms were constructed from single-molecule photon bursts identified as described above. At 165 mM ionic strength, where only a single transfer efficiency peak is visible due to fast exchange between free and bound conformations of ProTα, each histogram was fit with a Gaussian peak function to extract its mean transfer efficiency, $\langle E \rangle$. Consequently, for titration experiments, the mean transfer efficiency, $\langle E \rangle$, as a function of the concentration of GD was fit with

$$\langle E \rangle = \Delta\langle E \rangle_{sat}\frac{[G]_{tot} + K_{D,app} + [P]_{tot} - \sqrt{\left([G]_{tot} + K_{D,app} + [P]_{tot}\right)^2 - 4[G]_{tot}[P]_{tot}}}{2[P]_{tot}} + \langle E \rangle_0 \quad (5)$$

to obtain the apparent dissociation constant, $K_{D,app}$. Here, $[G]_{tot}$ and $[P]_{tot}$ are the total concentrations of GD and ProTα (labeled plus unlabeled ProTα), respectively, $\langle E \rangle_0$ is the mean transfer efficiency of free ProTα, and $\Delta\langle E \rangle_{sat}$ is the difference in transfer efficiency between free ProTα and ProTα saturated with GD.

At low ionic strength, where $N$ transfer efficiency peaks could be distinguished, the histograms were fit with two or more Gaussian peak functions to quantify the relative areas of the bound and unbound subpopulations and the corresponding fractions of the individual species, $p_i$ (with $i$ indicating the number of GD molecules bound to a ProTα chain; $p_0$ is the population of free ProTα) as a function of the GD concentration. The population curves obtained ($p_i$ as a function of GD concentration, Supplementary Fig. S3) were then fit to quantify the individual dissociation constants ($K_{D_i}$) with equations obtained by solving the model

$$\begin{cases} K_{D_i} = \frac{[PG_{(i-1)}][G]}{[PG_i]} = \frac{p_{(i-1)}[G]}{p_i} & [PG_i] = p_i[P] \, and \, [PG_0] = [P] \\ \sum_{i=0}^{N} p_i = 1 & i = 1, \ldots, N \\ [G]_{tot} = [G] + \sum_{i=1}^{N} i p_i[P]_{tot} \end{cases} \quad (6)$$

where $[PG_i]$ is the concentration of the complex of ProTα with $i$ GD molecules bound, and $[G]$ and $[P]$ are the concentrations of free GD and free ProTα, respectively. Equation 6 was also implemented in the 1:4 binding model for the analysis of the binding affinity of ProTα to GD at 165 mM ionic strength. In the 1:4 binding model, the observed transfer efficiency as a function of the total concentration of GD can be described as a linear combination of the population-weighted transfer efficiency of free ProTα, $\langle E \rangle_0$, and the transfer efficiencies in the

complexes with up to four GD molecules, $\langle E \rangle_i$ (with $i = 0, \ldots, 4$):

$$\langle E \rangle = \sum_{i=0}^{4} p_i \langle E \rangle_i \tag{7}$$

Combining this model with the relative increase in transfer efficiency observed in the coarse-grained simulations, we can rewrite Eq. 7 using $\langle E \rangle_i = \alpha_i \Delta \langle E \rangle_{sat} + \langle E \rangle_0$ as

$$\langle E \rangle = \sum_{i=0}^{4} p_i \left( \alpha_i \Delta \langle E \rangle_{sat} + \langle E \rangle_0 \right) \tag{8}$$

Here, $\alpha_i$ is the relative increase in transfer efficiency observed in the coarse-grained simulations, normalized by the difference in transfer efficiency between the 1:4 complex and free ProTα:

$$\alpha_i = \frac{\langle E \rangle_i^{CG} - \langle E \rangle_0^{CG}}{\langle E \rangle_4^{CG} - \langle E \rangle_0^{CG}} \tag{9}$$

We note that the analysis procedure developed here based on the additional insights from experiments and simulations yields affinities for the 1:1 complex that differ by about an order of magnitude from the values reported previously[16], where the transfer efficiencies were analyzed in terms of two defined subpopulations rather than the mean transfer efficiency used here.

## Nanosecond fluorescence correlation spectroscopy experiments

Data for nanosecond fluorescence correlation spectroscopy[45,49] were acquired for free labeled ProTα and in the presence of 100 mM excess of GD or in a mixture of 100 mM GD and 100 mM unlabeled ProTα. Donor and acceptor fluorescence emission from the subpopulation in the transfer efficiency histogram corresponding to labeled ProTα with active donor and acceptor fluorophores was correlated with a binning of the lag times of 0.5 ns. To avoid the effects of detector dead times and after-pulsing on the correlation functions, the signal was recorded using four detectors (with two detectors each for donor and acceptor) and cross-correlated between detectors. Acceptor and donor autocorrelations and donor-acceptor cross-correlations were fitted over a time window of 1 μs with

$$g_{ij}(\tau) = 1 + c(1 + c_{ab}^{ij} e^{-\frac{|\tau|}{\tau_{ab}^{ij}}})(1 + c_{cd}^{ij} e^{-\frac{|\tau|}{\tau_{cd}^{ij}}})(1 + c_T^{ij} e^{-\frac{|\tau|}{\tau_T^{ij}}}) \text{ with } i,j = A,D \tag{10}$$

in which $i$ and $j$ correspond to donor and acceptor fluorescence emission ($i,j = A, D$); The amplitude $c$ depends on the mean number of molecules in the confocal volume and background; $c_{ab}$, $\tau_{ab}$, $c_{cd}$ and $\tau_{cd}$ are the amplitudes and time constants of photon antibunching and chain dynamics, respectively; and $c_T$ and $\tau_T$ refer to the triplet blinking component on the microsecond timescale. Parameters without indices $ij$ are treated as shared parameters in the global fits of the auto- and cross-correlation functions. Distance dynamics result in a characteristic pattern of the correlation functions based on donor and acceptor emission, with a positive amplitude in the autocorrelations ($c_{cd}^{AA}$, $c_{cd}^{DD}$) and a negative amplitude in the cross-correlation ($c_{cd}^{AD}$), but with identical decay times. $\tau_{cd}$ was converted to the reconfiguration time of the chain, $\tau_{rec}$, as previously described[81] by assuming that chain dynamics can be modelled as diffusive motion in a potential of mean force derived from the sampled inter-dye distance distribution $P(r)$. In the present case, we used the $P(r)$ derived from a modified version of the self-avoiding random walk polymer chain (SAW-$\nu$)[82], which describes the behavior of even

very expanded intrinsically disordered proteins well:

$$P(r) = A \frac{4\pi}{\sqrt{\langle R^2 \rangle}} \left( \frac{r}{\sqrt{\langle R^2 \rangle}} \right)^{2+(\gamma-1)/\nu} \exp\left[ -\alpha \left( \frac{r}{\sqrt{\langle R^2 \rangle}} \right)^{1/(1-\nu)} \right] \tag{11}$$

where $\sqrt{\langle R^2 \rangle} = bN^\nu$ is the root-mean-squared end-to-end distance, $\nu$ is the scaling exponent, $\gamma \approx 1.1615$, $N$ is the number of inter-dye amino acid residues, and $b$ is approximately equal to 0.55 nm for polypeptide chains. $A$ and $\alpha$ are determined by the conditions $\int_0^\infty P(r)dr = 1$ and $\int_0^\infty P(r)r^2 dr = \langle R^2 \rangle$. $\sqrt{\langle R^2 \rangle}$ of labeled ProTα free and bound to GD was obtained by numerically solving $\langle E \rangle = \int_0^\infty P(r)E(r)dr$ for $\sqrt{\langle R^2 \rangle}$, where $E(r) = 1/[1 + (\frac{r}{R_0})^6]$ relates the transfer efficiency to the inter-dye distance $r$ ($R_0$ is the Förster radius).

The values of $\tau_T^{ij}$ for the acceptor and donor dyes were quantified by analyzing the correlation functions $g_{ij}(\tau)$ computed with logarithmically spaced lag times ranging from nanoseconds to seconds and fitted with:

$$g_{ij}(\tau) = 1 + c \frac{\left(1 + c_{ab}^{ij} e^{-\frac{|\tau|}{\tau_{ab}^{ij}}}\right) \left(1 + c_{cd}^{ij} e^{-\frac{|\tau|}{\tau_{cd}^{ij}}}\right) \left(1 + c_T^{ij} e^{-\frac{|\tau|}{\tau_T^{ij}}}\right) \left(1 + c_{T_2}^{ij} e^{-\frac{|\tau|}{\tau_{T_2}^{ij}}}\right)}{\left(1 + \frac{|\tau|}{\tau_D}\right) \left(1 + \frac{|\tau|}{s^2 \tau_D}\right)^{\frac{1}{2}}} \text{ with } i,j = A,D \tag{12}$$

where $\tau_D$ is the translational diffusion time of the labeled molecules through the confocal volume, $s$ is the ratio of the lateral to the axial radii of the confocal volume, and $c_{T_2}$ and $\tau_{T_2} > \tau_T$ are introduced to describe the observed multiexponential behavior of the donor and acceptor triplet times.

## NMR experiments

NMR experiments were acquired at 283 K on Bruker AVANCE III 600-, 750 MHZ ($^1$H) spectrometers with cryogenic probes or on a Bruker AVANCE NEO 800 MHz ($^1$H) spectrometer with cryogenic probe. Free induction decays were transformed and visualized using NMRpipe[83] or Topspin v. 3.7.0 or older (Bruker Biospin) and analyzed using CcpNmr Analysis software version 2.5[84]. All NMR samples were prepared in TBSK buffer (10 mM Tris, 155 mM KCl, 0.1 mM EDTA), 7.4 pH (at 283 K), 10% $D_2O$ (v/v), and 0.7 mM 4,4-dimethyl-4-silapentane-1-sulfonic acid (DSS). Proton chemical shifts were referenced internally to DSS at 0.00 ppm, and with heteronuclei referenced to their relative gyromagnetic ratios. For interaction studies in urea, the buffer additionally contained 4 M urea. Chemical shifts were transferred from BioMagResBank (BMRB) accession numbers 27215 (ProTα) and 34318 (GD).

$^1$H-$^{15}$N HSQC spectra were recorded on $^{15}$N ProTα (28 μM) in the absence and presence of equimolar addition GD, as well as with 8x molar ratio of RST and CaM separately, and on $^{15}$N-ProTα (37 μM) in the presence of 2x molar ratio of full-length H1. Furthermore, $^1$H-$^{15}$N HSQC spectra of $^{15}$N-ProTα (50 μM) were similarly recorded in the absence and presence of 4x molar ratio of the GD-variants. For the interaction studies in urea, $^1$H-$^{15}$N HSQC spectra of $^{15}$N-ProTα (25 μM) were acquired in the presence of either equimolar addition of H1 or 4x molar ratio of GD, both with and without 4 M urea. For NMR titrations, $^1$H-$^{15}$N HSQC spectra were acquired for $^{15}$N ProTα (28 μM) alone and added increasing concentration of GD up to 8x molar ratio and on $^{15}$N GD (54 μM) alone and added up to 3.32x molar ratio of ProTα. Before the titrations, the concentrated proteins were dialyzed in the same beaker. Subsequently, the isotope labelled protein sample was divided equally into two samples: one without ligand and one with the maximum concentration of ligand. $^1$H-$^{15}$N HSQC of the two samples were

recorded, obtaining titration endpoints. The titration points between these were obtained by sequentially adding the sample of maximum ligand concentration into the sample with no ligand.

The CSPs were calculated according to the following equation[85,86] (Eq. 13)

$$\Delta\delta_{NH} = \sqrt{(\Delta\delta H)^2 + (0.154^*\Delta\delta N)^2} \qquad (13)$$

The 'normalized CSP$_{sum}$' was calculated as the sum of the CSPs induced upon titration of $^{15}$N-ProTα with the GD variant, divided by the sum of the CSPs induced upon titration of $^{15}$N-ProTα with the GD-WT at the same stoichiometric ratio and performed on the same batch. I.e., a normalized CSP$_{sum}$ equal to 1 represent the same amplitude of CSPs induced, while a normalized CSP$_{sum}$ < 1 means lower amplitude and CSP$_{sum}$ > 1 a higher amplitude. The error was propagated from the error observed over 3 independent titrations of $^{15}$N-ProTα with the GD-WT.

Assignments of bound $^{13}$C,$^{15}$N ProTα (100 μM) at 1x and 8x molar ratio of GD WT were obtained from manual analysis of $^1$H-$^{15}$N HSQC and HNCACB spectra using CCPN analysis. SCS were calculated for each state using the SBiNLab random coil reference set for IDPs[87]. Assignments of free ProTα were obtained from previous work (BMRB ID: 27215)[16]. Backbone dynamics were assessed through $^{15}$N spin relaxation experiments determining $R_1$ and $R_2$ relaxation rates, and ($^1$H) $^{15}$N heteronuclear NOE experiments (hetNOE) using the pulse sequences hsqct2etfgpsi3d, hsqct1etf3gpsi3d and hsqcnoef3gpsi, respectively, provided by the Bruker BioPack. These experiments were recorded either on $^{15}$N ProTα (37 μM) in the presence of 74 μM full-length H1 (1:2) or on $^{15}$N ProTα (100 μM) in the absence and presence of up to 4x molar ratio of H1-GD or H1-GD variant at 750 MHz using different relaxation delays of (20, 60, 100, 200, 400, 700, 1100, 1300, 1500) ms and (34, 68, 102, 136, 204, 271, 407, 475, 543) ms for $R_1$ and $R_2$, respectively, with triplicate measurements on the same sample used for extracting error bars. HetNOEs where measured using saturation of $^1$H for 6 s. Additionally, these experiments were recorded on $^{15}$N GD (185 μM) in the absence and presence of equimolar addition of ProTα or on $^{15}$N GD (40 μM) in the presence of 0.25x ProTα, both at 600 MHz using different relaxation delays of (20, 60, 100, 200, 400, 600, 800, 1200) ms and (16, 32, 64, 128, 160, 192, 224, 256) ms for $R_1$ and $R_2$, respectively, with triplicate measurements on the same sample used for extracting error bars. HetNOEs where measured using saturation of $^1$H for 5 s. The obtained relaxation decays were fitted to a single exponential function and the relaxation times and hetNOEs determined using the CcpNmr Analysis software[84,88].

## All-atom molecular dynamics simulations

Molecular dynamics simulations described in the main text were performed using the DES-AMBER force field[51], employing the default version with residue net charges scaled by a factor of 0.9. Simulations were run of the ProTα complex with a single GD, of isolated ProTα and of isolated GD. For the complex and for ProTα by itself, a 17 nm truncated octahedron box was used. For the GD, a 6 nm truncated octahedron box was used. Two separate simulations of the complex were run, the first being initialized from a configuration in which a disordered ProTα was placed near the folded GD, while the second run was initialized from the configuration of the first simulation after 400 ns after reinitializing velocities with a different random seed. Simulations were conducted in explicit water using the TIP4P-D water model[51] in 165 mM sodium chloride. The simulations were run using GROMACS version 2018.3[89], with equations of motion integrated using a velocity Verlet algorithm with a time step of 2 fs and LINCS[90] constraints on all bonds. A constant temperature of 283 K was maintained using the Bussi velocity rescaling thermostat[91] with a 1 ps relaxation time, and constant pressure of 1 atm using the Parrinello-Rahman barostat[92] with a 5 ps relaxation time. Additional simulations of these

systems were also performed with different force fields. Further details of system composition and trajectory lengths are shown for all force fields in Supplementary Table 2, as well as on the zenodo repository: https://doi.org/10.5281/zenodo.11106958. Note that,"enhanced sampling" cannot be used for the present purposes, as essential dynamical properties are in general lost as a result of such techniques.

## Calculation of NMR observables from simulations

For each protein residue $i$, the trajectory of its backbone amide N-H vectors $\mathbf{r}_i(t) = \mathbf{r}_i^H(t) - \mathbf{r}_i^N(t)$ was first computed from the all-atom trajectory saved at 5 ps intervals, where $\mathbf{r}_i^H(t)$ and $\mathbf{r}_i^N(t)$ are, respectively, the positions of the amide hydrogen and nitrogen atoms. The correlation function

$$C_i(t) = \langle P_2(\boldsymbol{\mu}_i(t) \cdot \boldsymbol{\mu}_i(0)) \rangle, \qquad (14)$$

was calculated, where $P_2$ is the second Legendre polynomial, $P_2(x) = \frac{1}{2}(3x^2 - 1)$, and $\boldsymbol{\mu}_i(t) = \frac{\mathbf{r}_i(t)}{|\mathbf{r}_i(t)|}$. Relaxation rates were obtained from the spectral density of $C_i(t)$,

$$J_i(\omega) = 2 \int_0^\infty C_i(t) \cos \omega t dt \qquad (15)$$

In practice, the Fourier transform was performed by fitting a triple exponential to $C_i(t)$ and using the analytical transform of the fitted function. Relaxation rates $R_1$ and $R_2$ and steady-state NOEs, $\eta$, were given by:

$$R_1 = D(J(\omega_H - \omega_N) + 3J(\omega_N) + 6J(\omega_H + \omega_N)) + CJ(\omega_N) \qquad (16)$$

$$R_2 = \frac{D}{2}(4J(0) + J(\omega_H - \omega_N) + 3J(\omega_N) + 6J(\omega_H) + 6J(\omega_H + \omega_N)) + \frac{1}{6}C(4J(0) + 3J(\omega_N)) \qquad (17)$$

$$\eta = 1 + D\left(\frac{\gamma_H}{\gamma_N}\right)R_1^{-1}(6J(\omega_H + \omega_N) - J(\omega_H - \omega_N)) \qquad (18)$$

Where:

$$D = \frac{1}{20}\frac{(\mu_0/4\pi)^2\hbar^2\gamma_H^2\gamma_N^2}{r_{NH}^6} \qquad (19)$$

$$C = \frac{1}{15}\omega_N^2\Delta_{CSA}^2 \qquad (20)$$

In which $\hbar = \frac{h}{2\pi}$, $h$ is Planck's constant, $\mu_0$ is the vacuum magnetic permeability, $\gamma_H$ and $\gamma_N$ are the gyromagnetic ratios of $^1$H and $^{15}$N, respectively, $r_{NH}$ is the length of the amide N-H bond (0.1041 nm[93]), $\Delta_{CSA}$ is the chemical shift anisotropy (−170 ppm), and $\omega_H$ and $\omega_N$ are, respectively, the Larmor frequencies of the $^1$H and $^{15}$N nuclei at the magnetic field of interest.

To estimate errors, the trajectories were divided into $N = 10$ equal, non-overlapping windows, and the NMR observables $O_i$ were computed from each. The reported values are the mean of each observable over the different blocks, with the errors estimated as

$$\sigma_M[O_i] = \left(\frac{\langle O_i^2 \rangle - \langle O_i \rangle^2}{N}\right)^{1/2} \qquad (21)$$

To check the effect of initial conditions, two separate simulations with the DES-Amber force field, started from different initial structures and random seeds for velocities and thermostat, were analyzed. The

results (Supplementary Fig. 21) were consistent with each other considering the statistical error involved.

## Calculation of contact populations and contact lifetimes

Contacts were defined using a dual-cutoff scheme, in which a contact between two residues was initially formed if any two heavy atoms, one from each residue, came within a cut-off of 0.38 nm. If the closest distance between any two atoms, one from each residue, went above 0.8 nm, the contact was defined to be broken. Contacts between residues separated by fewer than 4 residues in sequence were not considered.

## Coarse-grained simulations of 1:1 binding

To elucidate the effect of variations in surface charge patterning on binding, a coarse-grained model was used. The model for wild-type GD was the same as used in ref. 16 except that residues of H1 not in the GD were deleted. The model for the mutants differed only in that an integer charge was assigned to each residue to match the residue charges in that mutant. Free energies of association were determined by umbrella sampling using 28 umbrella windows whose centers were equally spaced 0.5 nm apart between 0 and 2.5 nm, and 1 nm apart between 2.5 nm and 24.5 nm. A harmonic potential with a spring constant of 10 kJ/mol/nm$^2$ was used for all umbrellas, implemented using the GROMACS pull code, and reconstruction of potentials of mean force was done using WHAM[94]. The dissociation constant was determined by integrating the PMF[16]. The custom pair potential used in the model[16] is implemented in a modified version of gromacs available at: https://github.com/bestlab/gromacs-2019.4-cg

## Coarse-grained simulations of multiple GDs interacting with the single prothymosin α

The simulation parameters were identical to those used to simulate 1:1 binding (see above). We performed the following simulations: (i) Simulations with one prothymosin α and one to seven GDs in a 35 nm cubic box. Each of the 7 simulation setups consisted of 10 independent runs. The total length of each of the 7 setups was 30 μs. The first 0.5 μs of each run were treated as system equilibration and omitted from the analysis. (ii) Simulations containing a prothymosin α molecule with 20 GDs in a 70 nm cubic box (corresponding to the 100 μM concentration of GDs). We performed 6 independent runs with a total length of 20.7 μs. The first 0.3 μs of each run were treated as system equilibration and omitted from the analysis. In addition, we performed 18 independent runs of ProTα. Each of the runs was 5 μs long, and 0.3 μs of each run were treated as system equilibration and omitted from the analysis.

We determined the number of GDs interacting with prothymosin α by calculating the minimum distance between prothymosin α and each GD every 100 ps. The GD was considered to interact with prothymosin α if the minimum distance between them was less than 1.3 nm. Mean transfer efficiencies, ⟨E⟩, of the prothymosin α chain were obtained by calculating the instantaneous transfer efficiencies with the Förster equation $E(r) = R_0^6/(R_0^6 + r^6)$. Subsequently, the instantaneous transfer efficiencies for the prothymosin α chain with the determined number of interacting GDs were averaged over the simulation length. ⟨E⟩ for the one to three GDs interacting with prothymosin α was determined by averaging independent runs of simulation setup (i), since the number of GDs interacting with prothymosin α was rarely below 4 in simulation setup (ii). ⟨E⟩ for the four to seven GDs interacting with prothymosin α was determined by averaging independent runs of simulation setups (i) as well as (ii). We found ⟨E⟩ values obtained from simulation setup (i) and (ii) to be identical within the standard error of the mean, and we reported the average of two values. ⟨E⟩ for the eight GDs interacting with prothymosin α was determined by averaging independent runs of simulation setup (ii). Since we simulated prothymosin α without explicit representation of the fluorophores, the interdye distance, r, was estimated from the

simulations via the formula $r = d((N+9)/N)^\nu$, where $d$ denotes the distance between the C$^\alpha$ atoms of the labeled residues (residues 58 and 112 of prothymosin α); $N$ denotes the sequence separation of the labeling sites; and the scaling exponent ν was set to 0.6 − we thus approximate the length of dyes and linkers by adding a total of nine additional effective residues[95]. $R_O$ was set to 5.4 nm.

## Reporting summary

Further information on research design is available in the Nature Portfolio Reporting Summary linked to this article.

## Data availability

Molecular simulation input files and trajectories generated for this study are provided on Zenodo at https://doi.org/10.5281/zenodo.11106958. PDB codes of previously published structures used in this study are 6HQ1, 5OAO and 1CLL. Reference NMR data used in this work are BMRB 27215 and BMRB 34318. Source Data are provided as a Source Data file. Source data are provided with this paper.

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

## Acknowledgements

We thank Signe A. Sjørup and Morten Evers Nielsen for skilled technical assistance, Andreas Prestel, Fabien Ferrage, and Kaare Teilum for NMR advice, Daniel Nettels for providing data analysis software and help with instrumentation, and Kresten Lindorff-Larsen and Daniel Nettels for valuable discussions. This work was made possible by the Novo Nordisk Foundation Challenge grant REPIN – *rethinking protein interactions* (#NNF18OC0033926 to B.B.K. and B.S.), the Independent Research Fond Denmark (grant no.: 9040-00164B to B.B.K.) and the Swiss National Science Foundation (grant no. 310030_197776 to B.S.). R.B.B. was supported by the Intramural Research Program of the National Institute of Diabetes and Digestive and Kidney Diseases of the National Institutes of Health. We thank Villumfonden and the Novo Nordisk foundation for support for NMR infrastructure. NMR data were in part recorded at cOpenNMR - an infrastructure facility funded by the Novo Nordisk Foundation (#NNF18OC0032996). We used the computational resources of Piz Daint and Eiger at the CSCS Swiss National Supercomputing Center, and of the National Institutes of Health HPC Biowulf cluster (http://hpc.nih.gov). Mass spectrometry was performed in part at the Functional Genomics Center Zurich.

## Author contributions

K.B., B.S., R.B., and B.B.K. conceived the study, K.B., J.H.M., C.B.F., and D.S. purified the proteins and measured stabilities. K.B. and F.S.B. designed and performed all NMR experiments with assistance on titration experiments from D.S.; K.B. and F.S.B. analyzed all NMR data, M.T.I., A.S. and R.B. performed and analyzed the computational data, A.S. and F.K. performed single-molecule measurements and analyzed the data. K.B., F.S.B., A.S., R.B., and B.B.K. made the figures. K.B., B.S., R.B., and B.B.K. supervised the study, K.B., A.S., F.S.B., B.S., R.B., and B.B.K. wrote the manuscript. K.B. and B.B.K. were responsible for the overall project management and co-supervised the research.

## Competing interests

The authors declare no competing interests.
