## [Transparent Peer Review file · Nature Communications]

Role of charges in a dynamic disordered complex between an IDP and a folded domain

Corresponding Author: Professor Birthe Kragelund

Version 0:

Reviewer comments:

Reviewer #1

(Remarks to the Author)

The manuscript discusses the role of electrostatic interactions in the binding between an intrinsically disordered protein (IDP) and a folded domain. The IDP in question is ProtA (negatively charged), and the folded domain is the GD domain, derived from the positively charged IDP H1. This study was inspired by recent research on the binding mechanism between H1 and ProtA. While the binding between H1 and ProtA involves two IDPs, the current study examines the interaction between an IDP and a folded domain, allowing for an exploration of the electrostatically governed binding mechanism when only one partner is disordered. Although this study may have less biological relevance than the wild-type H1-ProtA binding, it addresses an intriguing question. Moreover, the study is comprehensive and well-designed, examining various system variants and employing a range of techniques, including NMR, FRET, and both atomistic and coarse-grained simulations. The question posed is of interest, and the results are insightful. However, several issues require further elaboration.

Comments:

1. It is concluded that the binding between GD and ProtA may satisfy different stoichiometries. Could this suggest that the dissociation constant (KD) might depend on the concentration of GD?
2. The coarse-grained simulations were effectively used to shed light on the complex formed between ProtA and multiple GD domains. Do these simulations include counterions? How does the number of released counterions depend on the stoichiometry of the GD-ProtA complex? Does the relative contribution of configurational entropy and counterion release entropy vary with stoichiometry?
3. Figure 5F: Earlier studies have shown that the release of counterions during the binding of H1 and ProtA significantly contributes to the thermodynamics of their interaction. Despite this, the entropy of association in Figure 5A neglects the entropy contribution from counterion release and focuses solely on configurational entropy. Is there a justification for this? It should be clarified what ΔH and ΔS represent.
4. As stated in the introduction, "the role of the number and distribution of charges for binding and selectivity has not yet been experimentally addressed..." While I agree with this statement, very similar questions have been addressed computationally in very related systems. I believe the discussion could benefit from mentioning and discussing these studies. Additionally, other studies have explored the role of net charge and charge clustering in IDPs, which the authors might consider discussing to further elaborate on the influence of polyampholyte molecular details on function.
5. The discussion section includes a brief survey of systems involving interactions between polyampholytes and folded domains, including intramolecular binding. The authors might consider adding information on relevant systems, such as the interactions between D/E repeats and positively charged domains (<https://doi.org/10.1016/j.jmb.2022.167660>; <https://doi.org/10.1016/j.jmb.2021.167122>).
6. Compensation between ΔH and ΔS : In addition to the previous comment (whether the compensation is still valid when the entropy term includes the entropy of ions/solvent), I wonder if the thermodynamics of binding would be similar if the variants with different net charges and charge clustering were in ProtA instead of the GD domain. It seems plausible that charge clustering and net charge might have a more significant effect on IDPs compared to folded domains. If this is correct, the conclusions of this study might be sensitive to the nature of the variants used. The discussion should be extended to address this possibility. Addressing the effect of net charge and/or charge clustering between two polyampholytic IDPs has not been done experimentally, nevertheless, I believe that extending the discussion of these two parameters for other related system is valuable.
7. Figure 1D: Does the KD correspond to H1 or GD?
8. Figure 3D is unclear, and I'm unsure if the text adequately refers to it.

Reviewer #2

(Remarks to the Author)

In the reviewed manuscript, Bugge et al. probe the dynamic, electrostatic interactions between prothymosin alpha (an IDP) and the small, folded domain of H1. Overall, the premise of the paper is interesting. However, some conclusions are overstated and a more in-depth discussion of the charge mutants investigated would increase the interest of the paper. The impact of the paper would increase by investigating the GDs of the different H1 variants alluded to in the discussion and extending the studies to test the relevance of GD mutants within full-length H1 along with investigating charge distribution in the tails.

Major Comments:

1. The initial results section ("ProTa remains disordered in complex with a folded partner") is largely reiterated from a previously-published 2018 paper (ref 16) and does not substantially add to it.
 - a. Similar data to figures 1B, 1C, 1G, 1H was published in the 2018 paper along with the grey data in Fig. 1D-1F. In my reading, the novel data are the NMR backbone amide relaxation data (R1, R2, hetNOE) of ProTa in the absence and presence H1 GD.
 - b. The conclusion of the section that "The complex[es] form without the formation of secondary or tertiary structure, and without structurally well-defined interaction sites or fixed relative orientations of the two proteins." is overstated from the data shown. Amide chemical shifts of helices can fall within similar regions and cannot be ruled out without ¹³C chemical shift indexing (which they did in Borgia 2018 with 15N-ProTa in complex with full-length H1). It is unclear how the data presented rules out well-defined interaction sites or relative orientations. The effects on ps-ns timescale dynamics and CSPs are clearly larger on some regions (perhaps what the authors refer to as the acidic region) than others.
 - c. Can the authors comment on the validity of interpreting Bracken plots for IDPs?
2. Regarding stoichiometries of ProTa-H1 GD complexes:
 - a. Please clarify the discrepancy in apparent K_d determined via smFRET at 165mM ionic strength between Borgia 2018 (reported as 1.9μM) and the 17μM reported here.
 - b. I have concerns about using Lohman-Record theory in a system with variable oligomeric state that depends on salt concentrations. Can the authors justify the validity?
 - c. The detour to address the question of different binding stoichiometries at different salt concentrations distracts from the overall story of the paper and brings up more questions than it answers. Especially since the concept is not returned to with the charge mutants. Is it possible that different subsets of conformational ensembles could lead to the same P<E> histograms observed in Fig. S3A rather than different binding stoichiometries?
 - d. At 165mM ionic strength, it is clear from Fig. 2F that different binding stoichiometries cannot be distinguished via smFRET. However, the NMR data does appear to support the formation of larger complexes, at least at the concentrations used in NMR. More information can be gleaned from the NMR data. To me, the NMR data look like they are collected under stoichiometric conditions. CSPs from Borgia 2018 (ref 16) were collected at 100μM 15N-GD and seem to support a 1:1 stoichiometry. CSPs from Fig. 2C are from 54μM 15N-GD and seem to support a 1:2 stoichiometry. It would be worth plotting peak intensities from Fig. S2 as a function of [ProTa]. To me, it looks like there is a dramatic decrease in intensity above a 1:1 molar ratio, supporting the formation of higher-order species. It is conceivable that the CSPs would plateau at 1:1 even if the binding does not if the chemical environment is the same regardless of the binding stoichiometry.
 - e. The implications and relevance of the stoichiometry of the binding interaction should be discussed as this is out of the context of full-length H1.
3. Regarding CG simulations:
 - a. I am concerned with the conclusion that the simulations are in "very good agreement" with the R1 and R2 data. I would, at a minimum, like references that support these levels of variation are considered negligible when comparing simulation and experimentally determined values. Neither the magnitude nor the residue-to-residue variation are in good agreement for R1 or R2, with the differences being most extreme for R2. On pg 5, R2 increases from an average of 3.1/s to 3.5/s (worded in a confusing manner in the text) were described as 'modest' regarding Fig. 1E, but differences between simulation and experiment of >2/s in Fig. 3B are glossed over and interpreted as being in "very good agreement". This warrants more discussion. These comments also extend to the differences in Fig. S5, which are described as being in "reasonable agreement". In addition, I did not find that the differences between CSPs and simulation contacts in Fig. 3D were addressed.
 - b. Furthermore, the authors write that they selected the force field that best agreed with the experimental data out of five tested (Fig. S7). Some vary wildly from the experimental data regarding NMR relaxation parameters. It would be beneficial to include intermolecular residue-residue contact lifetimes for all forcefields and correlation time for fluctuations in the radius of gyration of ProTa in Table S2. The large variability between force fields and the discrepancy between the chosen force field and experimental data make me skeptical of the CG simulation data.
 - c. Overall, it is warranted to tone down the language as well as include a discussion of the differences and what to take from them. I find it hard to make the claim: "The combined simulation and experimental results thus suggest that ProTα stays in contact with GD by constantly breaking and forming different contacts, without any specific long-lived interface." with the concerns raised here. They are consistent with this model, but do not suggest it. Are there alternative models that also fit the data?
4. Regarding charge mutants:

- a. Description of mutant design rationale is insufficient in main text. (Some text in methods section seems better suited for the main text.)
- b. Additional analysis of CD spectra should be conducted to assess changes in secondary structure with mutations to support the claim that the mutations do not affect structure (rather than simply thermal unfolding curves at 222nm).
- c. How was the stoichiometry of the H1 GD charge mutants in complex with ProTa assessed? Is it clear that they bind with a 1:1 stoichiometry as stated?
- d. Plots of CSPs as a function of residue do not provide insight into resonance trajectories.
- e. The authors plot the apparent K_d vs. 'normalized Csum' in Fig. 5C and state that it "suggest[s] that the amplitude of the CSPs is a measure of affinity in this type of complex". What molar ratio was the 'normalized Csum' calculated at? It is unclear from methods, but figures S15-S17 suggest it was calculated at a 1:1 ratio. Comparing CSP amplitudes is perhaps of only minimal interest if binding is not saturated as it would be expected for a weaker binder to have smaller CSPs. To make a statement about the bound complex of different mutants, it must first be shown that the spectrum is from the fully-bound state. Were the titrations collected up to 1:8 molar ratio? Did the chemical shifts continue to be perturbed?
- f. In addition to net charge, the charge distribution should be analyzed in more detail (quantitatively) and the charge density of mutants should be calculated across the surface. Depictions of surface electrostatics would show the reader how charge clustering changes between mutants. A more in-depth analysis should be done and discussed regarding the relationship between affinity for ProTa, net charge, and charge distribution across all the mutants tested.
- g. What are the cutoffs used to determine whether a mutant increases or decreases apparent affinity or is "neutral" (a confusing term to use when also discussing electrostatics)? (Is it within the error? And what type of error is being reported for the apparent K_d values?) (I am confused by the statement that the apparent K_d of 2S1 consistent with the sum of the two single-swap variants. It seems well within the error to be considered the same as 73_34.)
- h. How does net charge and charge distribution affect the dynamics of the complex?
- i. Concerns regarding the CG simulations extend here, and I question the validity of extracting enthalpies and entropies. I suggest validating enthalpies experimentally using ITC, at least for a subset of mutants, to make claims regarding enthalpy and entropy of binding.

Minor Comments:

5. Clarity would be improved by consistently referring to H1 GD vs. full-length H1 instead of GD vs. H1 in places. At times, the writing seems to suggest that GD and H1 are distinct entities rather than one being a part of the other.
6. In the analysis of the ProTa-H1 GD interaction from the perspective of the H1 GD, it would be helpful to see a surface version of the CSPs plotted on the cartoon structure in Fig. 2E. A complementary display of surface electrostatics on side-by-side structures would also be beneficial.
7. Clarify timescales of dynamics referred to with different methods (namely, NMR and smFRET).
8. Regarding figures:
 - a. Revise figure legends for clarity. I suggest reading through figure legends again to ensure full agreement between figures and legend text and to fill in details for clarity.
 - b. Labeling molar ratios in the format of ProTa:GD 1:0, 1:2, etc provides for improved clarity over 2xGD, etc. (Is it two times the concentration of GD or twice as much GD as ProTa? This requires additional thought.)
 - c. CSPs written in scientific notation are unconventional.
 - d. Residues are labeled unconventionally (e.g. 44G).
 - e. Figure legends (including supplementary) should have sample concentrations.
9. Table S3 is better suited for the main text.
10. Was the term "stoichiometric ratio" or "molar ratio" intended when describing the "normalized Csum"?
11. On page 7, it was described that "a 1:1 binding stoichiometry was enforced by measuring increasing amounts of an equimolar ratio of ProT α and GD". Using a 1:1 molar ratio is not the same as enforcing binding stoichiometry, which cannot be because binding interactions are concentration dependent.

Reviewer #3

(Remarks to the Author)

The manuscript "Role of net charges and charge clustering in a dynamic disordered complex between an IDP and a folded protein domain" from Bugge et al. describes the electrostatic interaction of a polyanionic IDP (ProT α) with a small folded domain (GD, net charge +9), and addresses a relevant topic for IDP community.

The effects of net charge and charge surface clustering in the globular domain (GD) were experimentally evaluated by NMR and single-molecule FRET, and further complemented with Molecular Dynamic simulations. A systemic approach was implemented by engineering GD (globular partner) for changing the net charge and its surface clustering with 25 variants, and using controls to exclude changes in the overall GD structure/ stability. The self-consistency of smFRET and NMR data is remarkable -regarding the fast exchange regime and the correlation between apparent K_D (from smFRET) and CSP amplitudes (from NMR).

The manuscript is well written, and the experiments are well thought-out. The data appear to be carefully acquired and analyzed. I recommend for publication at Nature Communication considering the very relevant conclusions for the IDP field. I only have a few suggestions / comments:

1. The rationale for choosing the labeling positions 58 and 110 for smFRET should be described.
2. For the low ionic strength conditions, the mean transfer efficiency, $\langle E \rangle$, for each subpopulation in smFRET should be listed or included in Figure S3.
3. It is not completely clear for the low ionic strength conditions, how the authors assigned the distinct subpopulation in smFRET histograms to the #GM molecules in the complex. Is it only from coarse-grained simulations?
4. The authors should discuss how the structure of GD isolated (WT variant used here) compares with GD in context of the full-length H1 (with disordered tails).

Minor corrections:

- o Fig 2D- The legend for top, middle and bottom does not match the figure order (R1, R2 and HetNOE in the figure).
- o Figs. 4D, S10, S15, S16 and S17- The color code used in the figures and described in legends does not correspond.
- o Page 11- The model used for determining the affinities by smFRET for ProTa-GD net charge variants should be included here.
- o Page 19- References for “corrected for background, channel crosstalk, acceptor direct excitation, differences in quantum yields of the dyes, and detection efficiencies.” should be included.
- o Some Equations in “Materials and Methods” – nsFCS and Calculation of NMR observables from simulations - lack the numeration.
- o Page 20 – “...function of GD concentration, Fig. S2” is Fig. S3.
- o The font size of the insert legend in Figure S3C should be increased.
- o Page 21- Some references of nsFCS- Materials and Methods (Gopich, et al., 2009 and Zheng, et al., 2018) are not included in the final reference list.
- o Review the journal abbreviations on “References “.

Version 1:

Reviewer comments:

Reviewer #1

(Remarks to the Author)

My comments are fully addressed.

Reviewer #2

(Remarks to the Author)

Many of the concerns regarding clarity and precision of language have been addressed. Only a few remain:

- Please remove “quantitative” from “there is quantitative agreement of the magnitude and residue-to-residue variation of the R1, R2 and hetNOEs between simulation and experiment.”
- For Fig. 3B, could some of the differences between experiment and simulation not be addressed/discussed? For example, for R2 values in the region of residues ~50-80, the NMR-determined values are unchanged in residues ~50-60 upon adding 1xGD while the simulations seem to show a substantial increase, but then they seem to be in agreement in ~70-80.
- I still advocate for presenting CSPs in ppm (rather than $\times 10^{(-1)}$ ppm) to avoid unintentionally misleading the reader. I am sympathetic to space, but to my eye, this could be done without altering the size of any plots.
- With the addition of a new Fig. 1D, I believe figure referencing in the text of the second half of the figure needs to be adjusted.

I understand that some of my initial suggestions were beyond the scope of the paper. Perhaps putting more emphasis on the final two figures (and related data that is currently in supplemental) and the associated text would increase the impact that is stated in the title of the manuscript.

Reviewer #3

(Remarks to the Author)

The authors addressed all my previous questions and comments.

This systematic work is focused on a relevant topic for the IDP community.

REVIEWER COMMENTS Nature Communications manuscript NCOMMS-24-39059

Reviewer #1 (Remarks to the Author):

The manuscript discusses the role of electrostatic interactions in the binding between an intrinsically disordered protein (IDP) and a folded domain. The IDP in question is Prota (negatively charged), and the folded domain is the GD domain, derived from the positively charged IDP H1. This study was inspired by recent research on the binding mechanism between H1 and Prota. While the binding between H1 and Prota involves two IDPs, the current study examines the interaction between an IDP and a folded domain, allowing for an exploration of the electrostatically governed binding mechanism when only one partner is disordered. Although this study may have less biological relevance than the wild-type H1-Prota binding, it addresses an intriguing question. Moreover, the study is comprehensive and well-designed, examining various system variants and employing a range of techniques, including NMR, FRET, and both atomistic and coarse-grained simulations. The question posed is of interest, and the results are insightful. However, several issues require further elaboration.

We thank the referee for his/her positive comments.

Comments:

1. It is concluded that the binding between GD and Prota may satisfy different stoichiometries. Could this suggest that the dissociation constant (K_D) might depend on the concentration of GD?

Indeed, if multiple GDs interact with a single Prota, the affinity is expected to decrease for each additional copy of GD, since the interactions with Prota need to be “shared” among several binding partners. At low salt concentration, the corresponding K_D values can in fact be resolved in single-molecule measurements, as illustrated in Fig. S3. We now emphasize this aspect more explicitly on page 5:

“Overall, this result strongly suggests that multiple GD molecules can bind to one Prota chain, with decreasing affinity for higher oligomers because of the anticooperativity resulting from the smaller number of charged groups in Prota effectively available per copy of GD bound (Fig. S3).”

2. The coarse-grained simulations were effectively used to shed light on the complex formed between Prota and multiple GD domains. Do these simulations include counterions? How does the number of released counterions depend on the stoichiometry of the GD-Prota complex? Does the relative contribution of configurational entropy and counterion release entropy vary with stoichiometry?

The coarse-grained simulations do not include explicit counterions, and the question of counterion release is investigated experimentally. Nonetheless, we note that even without

explicit counterions, the same model was able to reproduce the FRET efficiencies of the H1:ProTα complex (Borgia et al., 2018), suggesting that it can accurately capture the structure of these disordered complexes.

In the manuscript on p.6, we now included the following statement to emphasize this point: “Note that the coarse-grained simulations here did not explicitly consider counterions. While counterion release is an important phenomenon in the binding of charged proteins (Ref 46), we have shown that this model, which considers ions implicitly via screening of coulombic interactions, is sufficient to reproduce the structural ensembles of bound complexes of charged proteins (Ref 16).”

3. Figure 5F: Earlier studies have shown that the release of counterions during the binding of H1 and ProTα significantly contributes to the thermodynamics of their interaction. Despite this, the entropy of association in Figure 5A (we believe the reviewer means 5F?) neglects the entropy contribution from counterion release and focuses solely on configurational entropy. Is there a justification for this? It should be clarified what ΔH and ΔS represent.

Yes, indeed Figure 5F neglects counterion release, and this may conceivably vary with the “patchiness” of the charges on the GD surface. However, the fact that the model captures much of the trend in the experiment suggests that, at least to first order, it isn’t necessary to explicitly include counterion release. The reason for choosing this model was to focus deliberately on the ProTα configurational entropy, in order to explore the idea that, by interacting with clustered charges on the GD, ProTα might sacrifice less configurational entropy on binding. However, this turns out not to be the case. Rather, ProTα is able to form stronger coulombic interactions with the GD when charges are localized, and these stronger interactions in fact incur a larger penalty in configurational entropy.

In the manuscript (p. 10), we now refer to the “protein contributions to the entropy ($\Delta S_{\text{protein}}$) and enthalpy ($\Delta H_{\text{protein}}$) of binding” to make it clear what contributions we are talking about and have changed the axis legends in Fig. 5F accordingly. We also clarify: “A feature of our model is that it neglects thermodynamic contributions from solvent and ions. We can therefore identify the change in entropy in the model as arising from the protein configurational entropy change on binding.”

4. As stated in the introduction, “the role of the number and distribution of charges for binding and selectivity has not yet been experimentally addressed...” While I agree with this statement, very similar questions have been addressed computationally in very related systems. I believe the discussion could benefit from mentioning and discussing these studies. Additionally, other studies have explored the role of net charge and charge clustering in IDPs, which the authors might consider discussing to further elaborate on the influence of polyampholyte molecular details on function.

We agree with the reviewer and have expanded on the discussion to include a paragraph on the role of charges for IDPs and for polyelectrolyte interactions. The inclusion of this paragraph was important and has contextualized our work even further. Thanks for suggesting this. The relevant addition now reads (p. 11):

“The role of net charge and charge patterning for the function of IDPs is gaining increased attention with computational approaches important for effectively screening several sequences with modulated net charge and charge patterning. Several studies have shown that increasing charge clustering promotes chain compaction(17,58–60), modulates properties of polyelectrolyte condensates leading to more extended conformations(61,62), steers intersegmental transfer-efficiency for DNA bound transcription factors(63) and affects binding affinities, shown for the fully disordered complex between full-length H1 and ProTα (16,51,61). From these mostly computational studies, it appears that charge densities in IDPs may determine properties of their protein complexes including condensates such as specificity, affinity, dynamics and shape. Here we experimentally reveal the magnitude of such modulations and show that surface charge density and charge patterning of a folded protein can also affect these properties in complex with a charged IDP.”

5. The discussion section includes a brief survey of systems involving interactions between polyampholytes and folded domains, including intramolecular binding. The authors might consider adding information on relevant systems, such as the interactions between D/E repeats and positively charged domains (<https://doi.org/10.1016/j.jmb.2022.167660>; <https://doi.org/10.1016/j.jmb.2021.167122>).

We have included a suggested reference (p. 12) and the text now reads

“Stronger intramolecular interactions have also been observed, as exemplified by the highly dynamic, but high-affinity interaction between D/E repeats and the folded domain within HMBG1 leading to dynamic autoinhibition (ref 75)”.

6. Compensation between ΔH and ΔS : In addition to the previous comment (whether the compensation is still valid when the entropy term includes the entropy of ions/solvent), I wonder if the thermodynamics of binding would be similar if the variants with different net charges and charge clustering were in Prota instead of the GD domain. It seems plausible that charge clustering and net charge might have a more significant effect on IDPs compared to folded domains. If this is correct, the conclusions of this study might be sensitive to the nature of the variants used. The discussion should be extended to address this possibility. Addressing the effect of net charge and/or charge clustering between two polyampholytic IDPs has not been done experimentally, nevertheless, I believe that extending the discussion of these two parameters for other related system is valuable.

We agree that this would be an interesting question to investigate in future work, and we have now mentioned it in the discussion (p.12):

“Whether there is a corresponding effect of charge clustering in the disordered ProTa on binding is an interesting avenue for future investigation.”

7. Figure 1D: Does the KD correspond to H1 or GD?

We assume the reviewer is referring to Fig. 1J (former Fig. 1I), and we have clarified in the figure caption that this refers to GD.

8. Figure 3D is unclear, and I'm unsure if the text adequately refers to it.

We apologize – 3D was indeed not discussed in the text, and the previous reference to Fig. 3D should have been to Fig. 3C. We have corrected that reference, and added the following discussion of Fig. 3D (p. 7):

“As a final validation of the all-atom simulation results, we have compared the number of residue-residue contacts formed by each residue of ProTa with chemical shift perturbations on binding, both of which reflect the regions of ProTa to which the GD most frequently binds (**Fig. 3D**). The qualitative consistency of these two measures suggests that the GD is binding in the same region in the simulations as in the experiment.”

Reviewer #2 (Remarks to the Author):

In the reviewed manuscript, Bugge et al. probe the dynamic, electrostatic interactions between prothymosin alpha (an IDP) and the small, folded domain of H1. Overall, the premise of the paper is interesting. However, some conclusions are over-stated and a more in-depth discussion of the charge mutants investigated would increase the interest of the paper. The impact of the paper would increase by investigating the GDs of the different H1 variants alluded to in the discussion and extending the studies to test the relevance of GD mutants within full-length H1 along with investigating charge distribution in the tails.

We thank the reviewer for acknowledging the underlying premise of the paper and for the good suggestions for improvements provided below. The idea of investigating GDs from different H1 variants is interesting, but outside main message of this work. For the systematic studies conducted in present work, the variances in sequence by these GDs would add additional parameters to the study, complicating the analyses, and the approach is not essential for the conclusion of the work. Investigation of the charge distribution in the disordered tails is in our opinion a separate study outside the scope of the interplay between disordered and structured surfaces investigated here.

Major Comments:

1. The initial results section (“ProTa remains disordered in complex with a folded partner”) is largely reiterated from a previously published 2018 paper (ref 16) and does not substantially add to it.

The reviewer is correct that a small fraction of the data (Fig. 1c, b and h) is similar to those from our previous paper (Borgia et al., 2018), as stated in the figure legend. They are however, not identical as we have repeated the data using the molar ratios generally used in the present work, because it is helpful for providing sufficient context for the results of the current study. To fully characterize the interaction between ProTα and the folded GD - which is the focus of this work - we added substantially more data to fully cover the nature of the interaction, including dynamics and salt-dependence. Hence, we respectfully disagree that this part “does not substantially add to it” and argue that it is necessary to recapitulate this data to allow the reader to evaluate and follow this work without having to go the supplementary of another paper.

a. Similar data to figures 1B, 1C, 1G, 1H was published in the 2018 paper along with the grey data in Fig. 1D-1F. In my reading, the novel data are the NMR backbone amide relaxation data (R1, R2, hetNOE) of ProTα in the absence and presence H1 GD.

While Fig. 1c, b and h are repetition of data presented in a previous paper, where it was part of the supplemental data, we believe that these data are necessary for the independent understanding of the current work. For example, it is valuable for the interpretation of the relaxation perturbations to be able to compare to the CSPs. We do not claim that Fig. 1c, b and h are new findings, and reference is given in the figure legend to the previous paper.

b. The conclusion of the section that “The complex[es] form without the formation of secondary or tertiary structure, and without structurally well-defined interaction sites or fixed relative orientations of the two proteins.” is overstated from the data shown. Amide chemical shifts of helices can fall within similar regions and cannot be ruled out without ¹³C chemical shift indexing (which they did in Borgia 2018 with ¹⁵N-ProTα in complex with full-length H1). It is unclear how the data presented rules out well-defined interaction sites or relative orientations. The effects on ps-ns timescale dynamics and CSPs are clearly larger on some regions (perhaps what the authors refer to as the acidic region) than others.

We thank the reviewer for this suggestion. This aspect is relevant to pursue, and we have now assigned the carbon backbone chemical shifts for ProTα in its GD bound states (at 1:1 and 1:8 molar ratios of ProTα to GD) and have extracted the secondary chemical shifts (SCSs). As expected, the SCSs are similar, also to those of ProTα in the free and in the full-length H1 bound state (see Borgia et al., 2018). The data supports the conclusion that ProTα remains disordered in its complex with GD without induction of secondary structure. We have included these data in a new supplemental Fig. S1 and mention the data on p. 4:

“Finally, to assess whether binding to GD induces the formation of secondary structure in ProTα, we assigned the ¹³C-chemical shifts of the backbone nuclei of ProTα at a 1:1 molar ratio of GD and at full saturation with GD (**Fig. S1**). The secondary chemical shifts (SCSs) were unperturbed by binding of GD, as was the case for full-length H1¹⁶, underscoring the absence of structure induced in ProTα.”

c. Can the authors comment on the validity of interpreting Bracken plots for IDPs?

We thank the reviewer for raising this concern. We have discussed the use of Bracken plots for IDPs with experts in NMR relaxation. What is clear is that the limiting case of slow tumbling upon which the Bracken plot is based is not appropriate for IDPs. In folded proteins, for rigid regions, ~90% of the spectral density function decays with a τ_c of several nanoseconds. In IDPs, it's down to about 10% with a correlation time of several nanoseconds plus 50% at ~1 ns and 40% at ~50 ps. For the latter two timescales, we are not in the slow tumbling regime. In particular, the decay of the spectral density function between 0 and ω_N is limited. That does not mean the Bracken plot cannot be used for IDPs, but more elaborate approaches should be validated, which is far beyond the scope of the present work.

Based on these considerations, we have removed the Bracken plot of the relaxation data of ProTa from the supplemental information as well as the associated part in the results section (p. 4). Its removal has no influence on the conclusions of the work.

2. Regarding stoichiometries of ProTa-H1 GD complexes:

a. Please clarify the discrepancy in apparent K_d determined via smFRET at 165mM ionic strength between Borgia 2018 (reported as 1.9 μ M) and the 17 μ M reported here.

We thank the reviewer for pointing out this discrepancy, which may indeed cause some confusion. The main origin of the discrepancy is that in 2018, we had analyzed the data under the assumption that the binding reaction would be in slow exchange and involved only the 1:1 complex. Correspondingly, we used a different fit procedure, where two peaks of fixed transfer efficiencies but different populations were fit to the transfer efficiency histograms. If we apply this procedure to the data presented in the present manuscript, we obtain an affinity of $4 \pm 2 \mu$ M, close to the value we reported in 2018. In view of the additional insights regarding the binding mechanism from the current work, we now fit the average transfer efficiency as a function of GD concentration with suitable binding isotherms, as described in the Methods section. An additional factor that may contribute slightly to the apparent discrepancy is that, for preparative reasons, the GD variant used here differs by a terminal Gly residue. To prevent confusion, we now mention this aspect explicitly in the caption of Fig. 2 and provide more detail in the Methods section.

Figure 2 caption:

“We note that the analysis approach based on the insights developed here yields K_D values that are about an order of magnitude greater than previously reported (see Methods).”

Methods (p. 17):

“We note that the analysis procedure developed here based on the additional insights from experiments and simulations yields affinities for the 1:1 complex that differ by about an order of magnitude from the values reported previously, where the transfer efficiencies

were analyzed in terms of two defined subpopulations rather than the mean transfer efficiency used here.”

We would like to emphasize, however, that all measurements and analysis within this study were performed consistently, so that the relative effects observed between the different GD variants investigated are robust.

b. I have concerns about using Lohman-Record theory in a system with variable oligomeric state that depends on salt concentrations. Can the authors justify the validity?

Since we can separate the K_D values for the different oligomerization states at low salt (Fig. S3), and the K_D at higher salt is dominated by the 1:1 complex (Fig. 2F), the Lohman-Record analysis should still apply to good approximation, even if the distribution of oligomers populated changes with salt concentration.

c. The detour to address the question of different binding stoichiometries at different salt concentrations distracts from the overall story of the paper and brings up more questions than it answers. Especially since the concept is not returned to with the charge mutants. Is it possible that different subsets of conformational ensembles could lead to the same $P<E>$ histograms observed in Fig. S3A rather than different binding stoichiometries?

We appreciate the reviewer’s point that the salt concentration dependence adds a level of complexity. However, we see no way of omitting it, since it is the only approach, we have been able to identify that allows us to disentangle the contributions of the different stoichiometries. Only at low salt are the populations in slow exchange in the single-molecule FRET measurements, so that we can separate the subpopulations corresponding to different stoichiometries and extrapolate to higher salt concentrations. We have no reason to assume that the fundamental behavior as a function of salt concentration, especially the dominance of the first binding event for the change in transfer efficiency (Fig. 2G), differs between the WT GD and the GD variants, which is why we apply it to all variants. Our interpretation of the subpopulations in terms of different binding stoichiometries is supported by the separation of timescales between conformational dynamics within the complex and binding kinetics. The nsFCS data (Fig. S6) revealed chain dynamics in the complex on the timescale of tens of nanoseconds, without indications for slower contributions, much faster than the 1 ms time relevant for conformational averaging during the observation time in the confocal volume. Moreover, the different subpopulations are populated and depopulated systematically with increasing GD concentration, as expected for a system with multiple stoichiometries.

We now emphasize that the same procedure as implemented for the WT also accounts for the charge variants as long as we focus on the apparent affinities for the 1:1 complexes (p.8).

“The affinities of ProTα for the GD charge variants were quantified by smFRET (**Fig. S11**), where the reported apparent affinities are for the 1:1 complexes. These were inferred using the fitting procedures described above for the GD WT (**Fig. 2**) and the assumption that the fundamental behavior as a function of salt concentration, especially the dominance of the first binding event, do not differ between the GD WT and the GD variants.”

d. At 165mM ionic strength, it is clear from Fig. 2F that different binding stoichiometries cannot be distinguished via smFRET. However, the NMR data does appear to support the formation of larger complexes, at least at the concentrations used in NMR. More information can be gleaned from the NMR data. To me, the NMR data look like they are collected under stoichiometric conditions. CSPs from Borgia 2018 (ref 16) were collected at 100μM 15N-GD and seem to support a 1:1 stoichiometry. CSPs from Fig. 2C are from 54μM 15N-GD and seem to support a 1:2 stoichiometry. It would be worth plotting peak intensities from Fig. S2 as a function of [ProTα]. To me, it looks like there is a dramatic decrease in intensity above a 1:1 molar ratio, supporting the formation of higher-order species. It is conceivable that the CSPs would plateau at 1:1 even if the binding does not if the chemical environment is the same regardless of the binding stoichiometry.

Thank you for giving us the opportunity to clarify. At the conditions of the present work (54 μM) and with a K_D of 17 μM we are not working at concentration much larger than K_D and thus will not reach saturation at a stoichiometry of 1:1. In the Borgia et al. paper from 2018, we used a protein concentration closer to saturation, hence the differences.

The reviewer is correct that at higher molar ratios, higher order complexes will form, as also shown by smFRET. However, in these complexes ProTα stays dynamic, as can be inferred from the R2 values reported in Fig. 1E and the nsFCS data in Fig. S6. As we already mention in the main text, there are, as the reviewer noted, a few residues for which we observe peak broadening (Tyr28, Ala43, Lys69, and Ser71), not relating to any distinct patch on GD (p. 5). The effect is likely a result of approaching the intermediate exchange regime, as we mention on p. 4.

e. The implications and relevance of the stoichiometry of the binding interaction should be discussed as this is out of the context of full-length H1.

H1-GD in our studies represents a model of a folded protein with high surface charge, and it does not exist in isolation from full-length H1 in nature. However, as we speculate in the discussion, high affinity in polyelectrolyte complexes depends on the presence of high net charge per residue. To address the relevance of the propensity of modulation of stoichiometry in the case of a lack of charge matching between two interacting proteins, we have added a sentence to the discussion (p. 12):

“Finally, in the absence of charge matching between two polyelectrolytes, different stoichiometries, as we observe here, can occur, with differences of the corresponding

affinities. This behavior may be relevant in defining specificities between charged proteins and in their regulation.”

3. Regarding CG simulations:

a. I am concerned with the conclusion that the simulations are in “very good agreement” with the R1 and R2 data. I would, at a minimum, like references that support these levels of variation are considered negligible when comparing simulation and experimentally determined values. Neither the magnitude nor the residue-to-residue variation are in good agreement for R1 or R2, with the differences being most extreme for R2. On pg 5, R2 increases from an average of 3.1/s to 3.5/s (worded in a confusing manner in the text) were described as ‘modest’ regarding Fig. 1E, but differences between simulation and experiment of >2/s in Fig. 3B are glossed over and interpreted as being in “very good agreement”. This warrants more discussion. These comments also extend to the differences in Fig. S5, which are described as being in “reasonable agreement”. In addition, I did not find that the differences between CSPs and simulation contacts in Fig. 3D were addressed.

We would like to clarify that there were several sets of simulations performed here, some coarse-grained (CG) and some with all-atom force fields. The R1 and R2 data were computed from the all-atom simulations, as accurate dynamics would not be obtained from CG simulations. Regarding the agreement with experiment, this level of agreement represents the state of the art for IDPs and has never been achieved before without reweighting the simulation to match experiment. We note that there is significant statistical error associated with the simulated parameters because of the amount of simulation needed to sample binding/unbinding, and we have now included error bars for the simulations to reflect this. The key finding of the simulations is that, like the experiment, the changes in R1/R2 are modest. We have reworded the section on comparison with experiment as follows (p. 7):

“Overall, the results are in good agreement considering the difficulty of sampling these interactions in simulations: there is quantitative agreement of the magnitude and residue-to-residue variation of R1, R2 and hetNOEs between simulation and experiment. Just as important, the qualitatively small changes in these parameters in going from unbound to bound states are also reproduced in the simulations (Fig. 3B, C).”

(p. 8)

We have added some additional discussion of Fig. 3D as also requested by referee 1. The key feature is that binding, as probed by CSPs or by simulation contacts, occurs in the same region of ProTα. Obviously, a quantitative correlation between CSPs and simulation contacts is not expected.

b. Furthermore, the authors write that they selected the force field that best agreed with the experimental data out of five tested (Fig. S7). Some vary wildly from the experimental data regarding NMR relaxation parameters. It would be beneficial to include intermolecular

residue-residue contact lifetimes for all forcefields and correlation time for fluctuations in the radius of gyration of ProTα in Table S2. The large variability between force fields and the discrepancy between the chosen force field and experimental data make me skeptical of the CG simulation data.

To reiterate, none of these are CG simulations, they are all-atom simulations. It appears that the reason for the discrepancy in all but one of the force fields is due to salt bridges being too long-lived. To support this, we have computed contact lifetimes for all force fields as requested, and we show that all *except* the des-amber force field have a tail of very long-lived contacts (hundreds of nanoseconds) and updated Fig. S8 accordingly. The des-amber force field was specifically developed to avoid this artifact.

c. Overall, it is warranted to tone down the language as well as include a discussion of the differences and what to take from them. I find it hard to make the claim: “The combined simulation and experimental results thus suggest that ProTα stays in contact with GD by constantly breaking and forming different contacts, without any specific long-lived interface.” with the concerns raised here. They are consistent with this model, but do not suggest it. Are there alternative models that also fit the data?

We feel that our results actually support this conclusion strongly. The distribution of contact lifetimes from simulation with the des-amber force field shows that the contacts are short lived, consistent with this picture of not having a specific long-lived interface. The alternative is presumably that there is a specific long-lived interface. However, the simulation results with the other force fields that have long lived contacts shows how sensitive the NMR relaxation data are to such a scenario, suggesting that any long-lived interface would cause large changes in relaxation parameters on binding – which we do not observe in experiment. We have added the following discussion to emphasize this point (p. 7):

“This sensitivity of the relaxation parameters to the dynamics of the GD-ProTα interface supports our conclusion that the complex is stabilized by many short-lived interactions: if the alternative model is having long-lived contacts between the two proteins, we would expect to see strongly elevated R_2 values, which is not observed experimentally.”

4. Regarding charge mutants:

a. Description of mutant design rationale is insufficient in main text. (Some text in methods section seems better suited for the main text.)

We realize that we may not have been sufficiently explicit and have now added the following additional explanation (p. 9):

“The single-swap variants can be grouped in two; one group where charges were moved onto, or from, the folded α-helix 3 (α-variants), and one group where charges were moved

within the disordered and highly positively charged patch in the β -hairpin loop region (β -variants) (**Fig. 5A**).

To increase clarity of the used variants, we have also moved Table S3 into the main text as Table 1.

b. Additional analysis of CD spectra should be conducted to assess changes in secondary structure with mutations to support the claim that the mutations do not affect structure (rather than simply thermal unfolding curves at 222nm).

The stability of a folded protein is much more sensitive to mutation that changes in its structure, which was the rationale for including the measurements of the melting temperature (T_m) for each variant.

But in response to the reviewer's request, we have now included the far-UV CD spectra of the charge swap variants in Fig. S14B, which illustrate the similarity for all variants; CD spectra of the variants with changes in overall net charge have already been published in Martinsen et al., Protein Sci, 2022.

c. How was the stoichiometry of the H1 GD charge mutants in complex with ProT α assessed? Is it clear that they bind with a 1:1 stoichiometry as stated?

Even though the affinities of the different variants for ProT α differ, the affinities are generally expected to decrease with an increasing number of copies of GD bound (also similar to what was observed for the interaction of ProT α with full-length H1 in Sottini et al. 2020 and Chowdhury et al. 2023). Accordingly, the argument made in Fig. 2F regarding the dominance of the first binding event for the signal change and thus the measured K_D is expected to hold for all variants. Note that we emphasize the nature of the approximation and speak of an apparent K_D throughout. The uncertainty in the resulting K_D values is about a factor of two. However, given the broad range of affinities observed (Fig. 4E), our conclusions are robust. We now emphasize this point in the text on p. 8:

“The affinities of ProT α for the GD charge variants were quantified by smFRET (**Fig. S11**), where the reported apparent affinities are for the 1:1 complexes. These were inferred using the fitting procedures described above for the GD WT (**Fig. 2**) and assuming that the fundamental behavior as a function of salt concentration, especially the dominance of the first binding event, does not differ between the GD WT and the GD variants.”

d. Plots of CSPs as a function of residue do not provide insight into resonance trajectories.

We have now included examples to show that the variants follow the same chemical shift trajectory as the GD WT. These data are amended as Fig. S10B.

e. The authors plot the apparent K_d vs. ‘normalized Csum’ in Fig. 5C and state that it “suggest[s] that the amplitude of the CSPs is a measure of affinity in this type of complex”. What molar ratio was the ‘normalized Csum’ calculated at? It is unclear from methods, but figures S15-S17 suggest it was calculated at a 1:1 ratio. Comparing CSP amplitudes is perhaps of only minimal interest if binding is not saturated as it would be expected for a weaker binder to have smaller CSPs. To make a statement about the bound complex of different mutants, it must first be shown that the spectrum is from the fully-bound state. Were the titrations collected up to 1:8 molar ratio? Did the chemical shifts continue to be perturbed?

The CSPsum has been calculated at 4 times molar ratio of the GD variants, and hence at saturation. We thank the reviewer for pointing out that this was not clear from the figure legend, and this information has now been added to the legend of Fig. 5C. (p. 13)

f. In addition to net charge, the charge distribution should be analyzed in more detail (quantitatively) and the charge density of mutants should be calculated across the surface. Depictions of surface electrostatics would show the reader how charge clustering changes between mutants. A more in-depth analysis should be done and discussed regarding the relationship between affinity for ProTa, net charge, and charge distribution across all the mutants tested.

To our knowledge, there exist no single parameters that fully describe the surface electrostatics or the charge distribution of a folded protein and which can be calculated from a protein structure and correlated to the observed apparent K_D s. Thus, instead, we included the configurational entropy of the protein from the coarse-grained simulations as a measure to be correlated to the affinities.

As suggested by the reviewer, we have now included representations of the surface charge clustering for the GD variants and WT GD of Fig 5E and Fig 5F in a new supplemental figure S20. We mention this explicitly in the main text (p. 10).

g. What are the cutoffs used to determine whether a mutant increases or decreases apparent affinity or is “neutral” (a confusing term to use when also discussing electrostatics)? (Is it within the error? And what type of error is being reported for the apparent K_d values?) (I am confused by the statement that the apparent K_d of 2S1 consistent with the sum of the two single-swap variants. It seems well within the error to be considered the same as 73_34.)

We agree that the choice of wording was non-optimal and have changed “neutral” to “unchanged”.

The errors on the K_D s are standard errors from the fits, and we have now included this information. We selected mutants that are significantly different based on these errors.

The reviewer is correct, the effect of the double mutant 2S1 is not different to 73_34, because the effect of the second mutation (85_90) is unchanged compared to the WT.

h. How does net charge and charge distribution affect the dynamics of the complex?

This is an interesting question, and we have now addressed this using the four different GDs, the WT and three variants with alteration in charge distribution (74_70 and 74_67) and increased net charge (2E2Q), respectively. We have measured and compared the transverse relaxation rates R_2 s of ProT α in these complexes (4x molar excess of GD). We find that the dynamics of ProT α is maintained but is influenced by net charge and charge density. We have included these data in the manuscript in Fig. S20B and refer to the data in the main text as follows (p. 10):

“Finally, to address whether the changes in charge density or net charge of the GD would affect the dynamics of ProT α , we recorded ^{15}N R_2 relaxation rates of ProT α in complex with 4x molar excess of either GD WT or a GD charge variant. Here we selected two swap variants with increased (74_70) or decreased (74_67) charge density, respectively, and a variant with increased net charge (2E2Q). For all variants and similar to GD WT, we observed an increase in R_2 relaxation rates in the binding regions of ProT α and although small, the increase was more pronounced for the higher charge density variant 74_70 and the increased net charge variant 2E2Q (**Fig. S20B**). These data agree with the R_2 -profile of ProT α in complex with the more charged full-length H1¹⁶.

We thank the reviewer for stimulating this interesting additional insight.

i. Concerns regarding the CG simulations extend here, and I question the validity of extracting enthalpies and entropies. I suggest validating enthalpies experimentally using ITC, at least for a subset of mutants, to make claims regarding enthalpy and entropy of binding.

The model used to extract enthalpies and entropies is specifically focused on estimating configurational entropy, which can be straightforwardly identified from these CG simulations. On the other hand, it cannot be compared with experimental entropies which include other contributions. While CG simulations can be compared to dissociation free energies (K_D) at a single temperature (where they agree), their temperature dependence is expected to be different from experiment, and thus cannot be compared.

Minor Comments:

5. Clarity would be improved by consistently referring to H1 GD vs. full-length H1 instead of GD vs. H1 in places. At times, the writing seems to suggest that GD and H1 are distinct entities rather than one being a part of the other.

We agree that when the full-length H1 and the GD are mentioned together, we need to be concise. To avoid confusion, we have therefore changed the text to referring to H1-GD and full-length H1 when comparisons are made.

6. In the analysis of the ProTa-H1 GD interaction from the perspective of the H1 GD, it would be helpful to see a surface version of the CSPs plotted on the cartoon structure in Fig. 2E. A complementary display of surface electrostatics on side-by-side structures would also be beneficial.

We have changed figure 2E as suggested and now also show a surface representation of the electrostatics side by side the CSP plotted on the cartoon structure.

7. Clarify timescales of dynamics referred to with different methods (namely, NMR and smFRET).

We went through the text to make sure that the type of dynamics we discuss is either specified or obvious from the context.

8. Regarding figures:

a. Revise figure legends for clarity. I suggest reading through figure legends again to ensure full agreement between figures and legend text and to fill in details for clarity.

We thank the reviewer for pointing out that some information is incomplete — we have carefully gone through the figure legends and updated these accordingly for clarity.

b. Labeling molar ratios in the format of ProTa:GD 1:0, 1:2, etc provides for improved clarity over 2xGD, etc. (Is it two times the concentration of GD or twice as much GD as ProTa? This requires additional thought.)

We have updated the figure legends to clarify this issue.

c. CSPs written in scientific notation are unconventional.

We agree, but our notation is intentional to make the figure larger.

d. Residues are labeled unconventionally (e.g. 44G).

It is correct that the supplemental Fig S4, which is a direct export from the analyses program, showed this unconventional labeling. We have now changed this to the conventional format.

e. Figure legends (including supplementary) should have sample concentrations.

Thanks for noticing this omission. Sample concentrations are now included in the figure legends for main figures and supplementary figures, where this is appropriate.

9. Table S3 is better suited for the main text.

We have inserted Table S3 into the main text now, labelled as Table 1.

10. Was the term “stoichiometric ratio” or “molar ratio” intended when describing the “normalized Csum”?

It was recorded at 4 times excess, which has now been added to the legend of Fig. 5C.

11. On page 7, it was described that “a 1:1 binding stoichiometry was enforced by measuring increasing amounts of an equimolar ratio of ProTα and GD”. Using a 1:1 molar ratio is not the same as enforcing binding stoichiometry, which cannot be because binding interactions are concentration dependent.

We appreciate that the wording may have been suboptimal. Of course, different binding stoichiometries can still form in a 1:1 mixture, but performing the titration at a 1:1 concentration ratio will strongly decrease the probability of forming higher-order complexes compared to a titration where only one of the concentrations is changed. We now rephrased the corresponding sentence to read (p. 5):

“A similar analysis was conducted for stoichiometric titrations (blue points in Fig. 2F) at 165 mM ionic strength, where the contribution of stoichiometries with more than one GD molecule bound to ProTα was minimized by measuring increasing concentrations of an equimolar ratio of ProTα and GD.”

Reviewer #3 (Remarks to the Author):

The manuscript “Role of net charges and charge clustering in a dynamic disordered complex between an IDP and a folded protein domain” from Bugge et al. describes the electrostatic interaction of a polyanionic IDP (ProTα) with a small folded domain (GD, net charge +9), and addresses a relevant topic for IDP community.

The effects of net charge and charge surface clustering in the globular domain (GD) were experimentally evaluated by NMR and single-molecule FRET, and further complemented with Molecular Dynamic simulations. A systemic approach was implemented by engineering GD (globular partner) for changing the net charge and its surface clustering with 25 variants, and using controls to exclude changes in the overall GD structure/stability. The self-consistency of smFRET and NMR data is remarkable -regarding the fast exchange regime and the correlation between apparent KD (from smFRET) and CSP amplitudes (from NMR).

The manuscript is well written, and the experiments are well thought-out. The data appear to be carefully acquired and analyzed. I recommend for publication at Nature Communication considering the very relevant conclusions for the IDP field. I only have a few suggestions / comments:

We thank the reviewer for the positive comments to our work.

1. The rationale for choosing the labeling positions 58 and 110 for smFRET should be described.

Among the existing variants that have previously been tested for binding, this variant (which probes the more highly charged region of ProTα) has proven to exhibit the largest changes in transfer efficiency and is thus the most sensitive. We now provide more detail on this choice of variant in the Methods section (p. 13):

“Among the variants that have previously been tested for binding, ProTα E56C/D110C, which probes the more highly charged region of ProTα and was used here, has been shown to exhibit the largest changes in transfer efficiency and is thus the most sensitive to binding events”.

2. For the low ionic strength conditions, the mean transfer efficiency, $\langle E \rangle$, for each subpopulation in smFRET should be listed or included in Figure S3.

We have now included the mean transfer efficiency, $\langle E \rangle$ for each subpopulation in the smFRET as vertical dashed lines in Fig. S3.

3. It is not completely clear for the low ionic strength conditions, how the authors assigned the distinct subpopulation in smFRET histograms to the #GM molecules in the complex. Is it only from coarse-grained simulations?

Given the appearance and disappearance of subpopulations in the transfer efficiency histograms at low salt concentration, the hypothesis that they correspond to the different stoichiometries seemed plausible, also in view of our previous work on the different stoichiometries in the complexes between ProTα and full-length H1 (Chowdhury et al., PNAS 2023), where analogous behavior was observed. Moreover, the subpopulations observed with increasing GD concentration exhibit increasing transfer efficiency, as expected from a compaction of ProTα with an increasing number of copies of oppositely charged GD bound. The hypothesis was then further tested and confirmed with the coarse-grained simulations. In the revised version, we have rephrased our train of thought more explicitly, including the analogy to the previous work on full-length H1 (p. 5).

4. The authors should discuss how the structure of GD isolated (WT variant used here) compares with GD in context of the full-length H1 (with disordered tails).

This is an important point that we already addressed in a previous work on the structure determination of H1-GD (Martinsen et al., Protein Science 2022). We found no difference in T_m or the far-UV CD for GD when isolated compared to GD within the full-length H1. To make this point clearer to the reader, we have now updated the introduction in main text as follows (p. 3):

“When isolated, GD has similar properties in terms of structure and stability when isolated as in the context of full-length H1 (Ref 34)”.

We thank the reviewer for pointing out this was missing from the manuscript.

Minor corrections:

- o Fig 2D- The legend for top, middle and bottom does not match the figure order (R1, R2 and HetNOE in the figure).

Thanks for noticing this mistake, it has now been corrected.

- o Figs. 4D, S10, S15, S16 and S17- The color code used in the figures and described in legends does not correspond.

This has now been corrected.

- o Page 11- The model used for determining the affinities by smFRET for ProTa-GD net charge variants should be included here.

We now mention specifically that the K_D refers to the 1:1 complex (p. 8):

“The affinities of ProTa for the GD net charge variants were quantified by smFRET (Fig. S11), where we refer to the affinities of the 1:1 complexes, which were inferred as described above (Fig. 2).”

- o Page 19- References for “corrected for background, channel crosstalk, acceptor direct excitation, differences in quantum yields of the dyes, and detection efficiencies.” should be included.

Thank you for noticing this omission — a reference has now been included.

- o Some Equations in “Materials and Methods” – nsFCS and Calculation of NMR observables from simulations - lack the numeration.

The equations have now been numbered.

- o Page 20 – “...function of GD concentration, Fig. S2” is Fig. S3.

Thank you, has now been corrected.

o The font size of the insert legend in Figure S3C should be increased.

This has now been increased and should be better readable.

o Page 21- Some references of nsFCS- Materials and Methods (Gopich, et al., 2009 and Zheng, et al., 2018) are not included in the final reference list.

The two references are now included, thanks for catching this.

o Review the journal abbreviations on “References “.

Thank you, all references have been checked and their journal abbreviations updated.

COMMENTS TO REVIEWERS

Reviewer #1 (Remarks to the Author):

My comments are fully addressed.

Thank you

Reviewer #2 (Remarks to the Author):

Many of the concerns regarding clarity and precision of language have been addressed. Only a few remain:

- Please remove “quantitative” from “there is quantitative agreement of the magnitude and residue-to-residue variation of the R1, R2 and hetNOEs between simulation and experiment.”

“quantitative” has been removed from the sentence

- For Fig. 3B, could some of the differences between experiment and simulation not be addressed/discussed? For example, for R2 values in the region of residues ~50-80, the NMR-determined values are unchanged in residues ~50-60 upon adding 1xGD while the simulations seem to show a substantial increase, but then they seem to be in agreement in ~70-80.

We have expanded on the comparison to directly comment on the region between residues 50-80 as suggested by the reviewer as follows:

"Overall, the results are in ~~very~~ good agreement considering the difficulty of sampling these interactions in simulations: there is ~~quantitative~~ agreement of the magnitude and residue-to-residue variation of R1, R2 and hetNOEs between simulation and experiment. Just as important, the qualitatively small changes in these parameters in going from unbound to bound states are also reproduced in the simulations (Fig. 3B, C). ~~The regions in which there is the greatest disagreement with experiment, e.g. R2 for residues 50-80 with 1 GD bound, are also the regions with the largest statistical errors.~~ NMR relaxation parameters for GD were also in reasonable agreement with experiment (Fig. S5)."

- I still advocate for presenting CSPs in ppm (rather than $\times 10^{-1}$ ppm) to avoid unintentionally misleading the reader. I am sympathetic to space, but to my eye, this could be done without altering the size of any plots.

The reviewer has a point, and we do not intend to mislead the reader. Therefore, we have changed all figures accordingly to show the CSPs in ppm (rather than $\times 10^{-1}$ ppm). This include Fig. 1B, 1H; Fig. 2B, 2C; Fig. 3D; Fig. 4C, 4D, 4G + supplemental figures S10A-D, S15, S16.

- With the addition of a new Fig. 1D, I believe figure referencing in the text of the second half of the figure needs to be adjusted.

This is correct. Referencing has been updated, thank you for noticing this.

I understand that some of my initial suggestions were beyond the scope of the paper. Perhaps putting

more emphasis on the final two figures (and related data that is currently in supplemental) and the associated text would increase the impact that is stated in the title of the manuscript.

The emphasis on the data presented in the last two figures (Fig. 4 and Fig. 5) is already extensive and take up a large fraction of the result section as well as of the discussion. However, to tie in the new supplemental data on the relaxation of ProT α in the different GD complexes, we have added a sentence to the discussion as follows, highlighted in red.

“Thus, and in agreement with a recent computational study of charge variations of ProT α and H1⁵³, the relative position of charges matters. **The more positive charges that are present on the surface of GD, and the more clustered they are, the larger the loss in configurational entropy of ProT α in the complexes.** Importantly, our findings suggest that it is not the specific position of a charge, as it would be within a traditional binding site, it is its contribution to an overall local and global charge density that is relevant.”

Reviewer #3 (Remarks to the Author):

The authors addressed all my previous questions and comments.
This systematic work is focused on a relevant topic for the IDP community.

Thank you